# CRAFTIUM: A FRAMEWORK FOR CREATING SINGLE AND MULTI-AGENT ENVIRONMENTS FOR OPEN-ENDED AND EMBODIED AI

## ABSTRACT

Advancements in open-ended and embodied AI require highly adaptable and computationally efficient environments. Yet, existing platforms often lack the flexibility, efficiency, or richness necessary to drive progress in these areas. Research in fields related to open-endedness, such as unsupervised environment design and continual reinforcement learning, usually defaults to simplistic 2D environments, as alternatives are either too rigid or computationally expensive. Conversely, in embodied AI, the field relies on fully featured video games like Minecraft, which are rich in content but computationally inefficient, single-agent only, and hinder the creation of new tasks. This paper introduces Craftium, a framework based on the open-source Minetest game engine, providing a highly customizable, easy-to-use, and efficient platform for building rich single and multi-agent Minecraft-like 3D environments. We showcase environments of different complexity and nature: from single and multi-agent reinforcement learning tasks to vast worlds with many creatures and biomes and customizable procedural task generators. Conducted benchmarks show that Craftium substantially improves the computational cost of Minecraft-based frameworks, achieving +2K steps per second more.

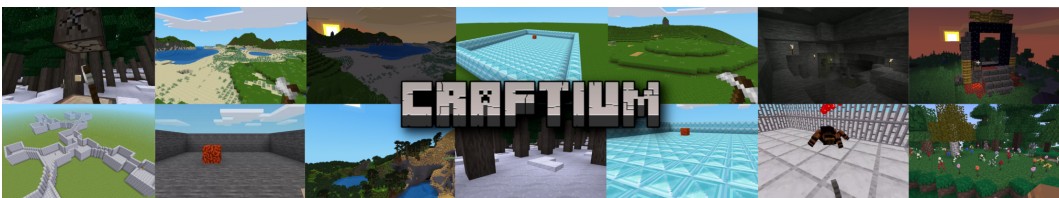

Figure 1: Examples of the diverse single and multi-agent environments that can be created in Craftium. From simple reach-the-goal tasks to battling hostile creatures, surviving in procedurally generated dungeons, and exploring vast open worlds filled with animals, monsters, and varied biomes.

## 1 INTRODUCTION

Progress in open-ended (Hughes et al., 2024) and embodied AI (Paolo et al., 2024), as well as in Reinforcement Learning (RL) (Sutton & Barto, 2018) is inherently tied to the environments where agents are trained, evaluated, and analyzed. Each new insight or advancement in the field is supported by an environment that enables its emergence and study. A well-known example is the Atari Learning Environments (ALE) (Bellemare et al., 2013), which undoubtedly contributed to the advancement of the RL field marking many of its most important milestones. To name a few: the introduction of the Deep Q-Networks (Mnih et al., 2013), the "infamously difficult Montezuma's Revenge" (Bellemare et al., 2016) that inspired many exploration strategies (Ostrovski et al., 2017; Burda et al., 2019; Badia et al., 2020b), and the first time an agent outperformed humans in all Atari benchmarks (Badia et al., 2020a).

However, the research in these areas is bound to challenges introduced by the employed environments, as largely observed throughout the literature. The researcher often faces a dilemma between

computationally efficient but simplistic, or substantially slower but rich environments. For instance, Continual Reinforcement Learning (CRL) (Abel et al., 2023), Unsupervised Environment Design (UED) (Garcin et al., 2024), and Multi-Agent RL (MARL) (Ying et al., 2023), are greatly affected by the efficiency of the employed environments as require learning from many tasks or agents. Thus, in these works, experiments are often limited to simple environments as a consequence of the computational cost of employing more complex alternatives (Rigter et al., 2024; Malagon et al., 2024; Beukman et al., 2024; Rutherford et al., 2024). For example, Craftax relies on 2D grids (Matthews et al., 2024), while OMNI-EPIC (Faldor et al., 2024) employs 3D environments of substantially limited diversity compared to alternatives like MineDojo (Fan et al., 2022) or Habitat 3.0 (Puig et al., 2024).

Conversely, works on rich and complex environments (Grbic et al., 2021; Earle et al., 2024; Prasanna et al., 2024; Raad et al., 2024) rely on fully featured video games that have a high computational cost and are close-source. The best-known of such platforms is Minecraft, which has inspired several single-agent environments and benchmarks over the years (Johnson et al., 2016; Guss et al., 2019; Fan et al., 2022). However, Minecraft is a fully featured and complex 3D game, which makes it substantially more inefficient than simpler alternatives (Wydmuch et al., 2019; Matthews et al., 2024). Furthermore, its closed source greatly limits its flexibility, hindering its application to problems beyond *classic* RL, like UED, CRL, and MARL scenarios.

Another important issue that specially affects research in these areas is the lack of flexibility of the environments. Commonly used environments offer no customization or limited possibilities often restricted to a set of predefined parameters, such as difficulty level or the number of enemies. Among others, these environments include: ALE (Machado et al., 2018), MineRL (Guss et al., 2019), ProcGen (Cobbe et al., 2020), MineDojo (Fan et al., 2022), Crafter (Hafner, 2022), and Craftax (Matthews et al., 2024). The lack of flexibility hinders the ability to analyze specific behavior of agents, obstructing algorithmic comparison beyond pure performance benchmarking, which has been shown insufficient for RL (Jordan et al., 2024). Although flexible platforms that allow the creation of new and diverse environments exist, these fall into 2D worlds (Bamford et al., 2020; Chevalier-Boisvert et al., 2023; Matthews et al., 2024) or depend on complex Domain Specific Languages (DSL) that difficult their implementation, while still not being 3D, as it is the case of VizDoom (Wydmuch et al., 2019) and MiniHack (Samvelyan et al., 2021).

In this paper, we present Craftium, a platform for easily creating rich 3D environments for single and multi-agent embodied and open-ended AI research, thought to be highly customizable and efficient. Unlike most complex environment platforms, that are based on video games (e.g., VizDoom is based on ZDoom and MiniHack in NetHack), Craftium is based on a game engine: Minetest (Minetest Team, 2024b). This allows easy creation of rich single and multi-agent voxel environments[1] using a powerful and greatly documented Lua API (Minetest Team, 2024a), instead of much less popular DSLs. Lua (Ierusalimschy, 2006) is a Python-like, easy-to-use and understand, mature, and efficient programming language used in many popular tools and projects (e.g., Roblox, World of Warcraft, and Neovim). In Craftium, Lua is used to expose a complete game engine (i.e., Minetest) to develop environments. Moreover, Minetest is open-source and has a vibrant community that has created many games and assets that can be used in Craftium environments (Ward, 2023a), significantly reducing the development cost of complex scenarios. For instance, all the environments shown in Figure 1 have been implemented in less than 160 lines of code, comments and whitespace included. These environments, later described in Section 14, showcase the versatility of the presented framework, from RL and MARL tasks of various levels of complexity and different nature, customizable procedural environment generators for UED, CRL, and meta-RL (Yu et al., 2020; Rimon et al., 2024) to gigantic procedurally generated open worlds ($64K \times 64K \times 64K$ blocks) for research on open-ended (Hughes et al., 2024) and embodied AI (Paolo et al., 2024). Beyond being flexible, feature-rich, and developer-friendly, we show that Craftium environments run $38\times$ faster than alternatives based on the original Minecraft game, the only platforms that offer a similar level of complexity and richness. Furthermore, Craftium implements the popular Gymnasium (Towers et al., 2024) and PettingZoo (Terry et al., 2021) interfaces the modern standard for RL and MARL research[2] respectively, making it compatible with many other libraries and projects (Raffin et al., 2021; Huang et al., 2022b; Serrano-Muñoz et al., 2023). Finally, Craftium is fully open source and

---

[1]Voxel games use 3D blocks (voxels) to construct and represent the game world, allowing players to modify the environment by adding or removing blocks. Figure 1 shows examples of these types of games.

[2]Although can be used for learning paradigms beyond RL (e.g., evolutionary algorithms).

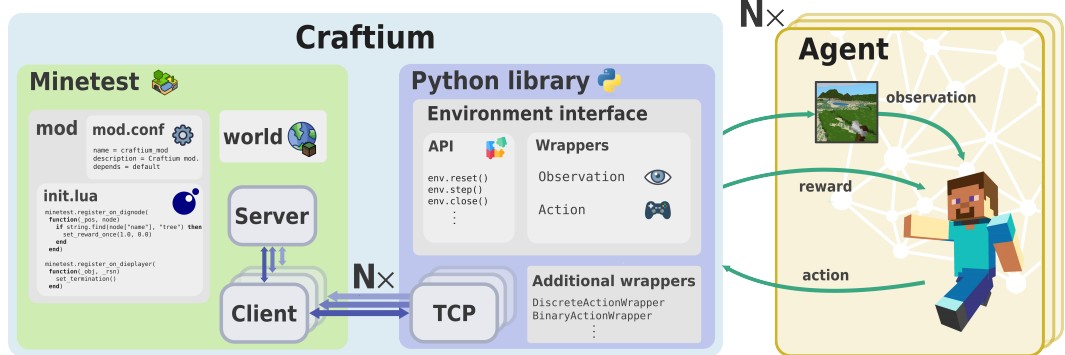

Figure 2: Overview of Craftium's internal architecture. Components denoted with $\times N$ are repeated according to the number of agents (one or more).

includes extensive online documentation with many guides, usage examples, tutorials, a detailed reference, and ready-to-use scripts.[3]

## 2 BACKGROUND: MINETEST AND MINECRAFT

Minecraft is a very popular (Gerken, 2023) sandbox game[4] where players explore a voxel-based, procedurally generated 3D world, gather resources, craft tools, and build structures. Although Minecraft supports partially extending the original game (user extensions are referred to as mods), this is limited by its close source nature and restrictions to access its underlying logic. Additionally, the game and its mods are implemented in Java, a programming language not tailored for high-performance applications and usually not commonly accessible in HPC clusters.

In contrast, Minetest is a voxel-based game engine inspired by Minecraft, serving as a platform for creating games rather than being a game itself. Unlike Minecraft, Minetest supports modding at its core, allowing fine-grained real-time access and modification of the internal state of the game engine. This enables extensive customization of its behavior, facilitating the creation, modification, and extension of existing games using its powerful Lua API (Minetest Team, 2024a; Ward, 2023b). In turn, Minetest is implemented in C++, a widely adopted programming language and known for its high efficiency. Moreover, Minetest is open source and is supported by an active community that has created hundreds of open free-to-use games and mods (Ward, 2023a), that are seamlessly loaded in Craftium (as employed in all environments from Section 3.5).

## 3 CRAFTIUM

Craftium follows the architecture illustrated in Figure 2. It consists of two main components: the Minetest game engine and the Python environment interface. This interface is the bridge between the environment and agents. Internally, it handles a communication channel per agent, which connects to Minetest, sending and receiving data such as observations, actions, or rewards. On the other hand, the Minetest server executes the logic of the environment, specified by a file characterizing the 3D world and a script (i.e., mod) that defines its behavior. The Minetest server also synchronizes the Minetest clients (one per agent), which handle rendering and communication tasks with the Python library. Finally, note that the original Minetest game engine does not support these features, but its open-source nature allowed modifying its code to support this architecture (see Appendix A for details).

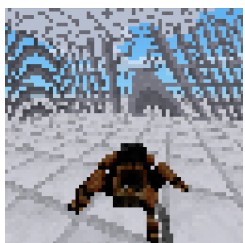

Figure 3: Example of a 64x64 pixel RGB image observation.

---

[3]A link to the online documentation will be provided upon acceptance.

[4]Sandbox games allow players extensive creative freedom to explore, build, and manipulate the game environment with few constraints or predetermined goals.

```
1 name = craftium_mod
2 description = My env.
3 depends = default
```

Figure 4: Example configuration file of a mod implementing a Craftium environment. This example depends on the `default` mod that provides several basic functionalities.

```
1 minetest.register_on_dignode(function(ps, block)
2   if string.find(block["name"], "tree") then
3     set_reward_once(1.0, 0.0)
4   end
5 end)
6
7 minetest.register_on_dieplayer(function(obj, rn)
8     set_termination()
9 end)
```

Figure 5: Lua script (i.e., mod) implementing basic environment mechanics.

In the following, Sections 3.1, 3.2, and 3.3 describe Craftium environments, the creation process, and the interface to use them. Respectively, Section 3.4 compares the performance of Craftium with other frameworks. Finally, Section 3.5 showcases the presented framework as a general-purpose environment creation tool across a variety of use cases and fields concerning autonomous agents.

### 3.1 OBSERVATIONS, ACTIONS, AND REWARDS

**Observations.** In Craftium, observations are images from the agent's point of view. An example observation is provided in Figure 3. Observations are highly customizable (e.g., size, number of channels, etc.) and can vary between environments. Moreover, Craftium supports many popular techniques such as, frame skipping and frame stacking that are commonly used throughout the literature (Huang et al., 2022a).

**Actions.** By default, actions are composed of a combination of 21 keyboard actions and a tuple that defines the movement of the mouse, which is mainly used to control the camera. Keyboard-related actions are binary variables with a value of 1 if the key is pressed, and 0 otherwise. The movement of the mouse is defined with the tuple $(\Delta_x, \Delta_y) \in [-1, 1]^2$, where $\Delta_x < 0$ moves the mouse to the left in the horizontal axis and $\Delta_x > 0$ to the right, similarly, $\Delta_y < 0$ moves the mouse downwards in the vertical axis and $\Delta_y > 0$ moves it upwards. Thus, if $\Delta_x = \Delta_y = 0$, the mouse is not moved. See Appendix B.1 for a detailed description of all the possible actions supported in Craftium.

The default action space is designed to be versatile, covering as many use cases as possible: from tasks with complex action sequences (e.g., manual inventory control) to simple navigation environments with a couple of actions (e.g., forward and lateral movement). However, the default action space is overly complex for most tasks: the number of possible keyboard action combinations in the default space is $2^{21}$. Therefore, Craftium allows reducing the action space to the minimal subset required to solve the task at hand substantially simplifying the learning process of the agent (see Appendix B.2).

**Rewards.** In Craftium, reward functions are defined using Lua scripts (mods are discussed in the next section). The framework provides a comprehensive set of tools for this purpose, including an extended version of the Minetest Lua API. This functionality is implemented in a modified version of the game engine developed specifically for this work, which incorporates additional functions for setting and retrieving reward values and episode termination flags. An example of these functions is presented in Section 3.2. A complete list of modifications to the original Minetest game engine can be found in Appendix A, while additional functions for defining RL environments are detailed in Appendix C.

### 3.2 CREATING CUSTOM ENVIRONMENTS

Creating a Craftium environment implies two steps: ① generating a *world*: a database with all the information about the virtual environment where the agent will be placed and will interact with (Figure 1 shows images of a variety of worlds); and ② defining the behavior of the environment,

such as the reward function and conditions for episode termination. The following lines describe these steps in detail.

① Minetest offers practically unlimited possibilities for generating worlds. However, creating a world can be as simple as a few clicks when using one of the many predefined map generators.[5] If finer control over the map generation process is needed, maps can be created using custom scripts. The procedural environment generator presented in Section 3.5.4 is an example of a complex custom map generation process.

② The next step is to define the behavior of the environment. This is done via mods: user-defined scripts that modify and extend the game engine's behavior, allowing for the creation of custom environments, mechanics, and interactions within the 3D world. A mod has a minimum of two files: a configuration file, and a Lua script.

The **configuration file** contains the mod's metadata. It commonly includes the mod's name, a description, and the list of dependencies (see Figure 4). The **Lua script** is where the environment's mechanics are implemented. Figure 5 illustrates an example script that defines the task of chopping as many trees as possible (presented in Section 3.5.1). Line 1 registers a *callback* function that is called every time the player (i.e., agent) digs a block. In line 2, this function checks if the dug block is part of a tree; if the condition is met, line 3 sets the reward to 1 for that timestep (`set_reward_once` and other RL related functions are described in Appendix C). Line 7 registers another callback function. In this case, the function is run every time the player dies and calls another function that terminates the episode, in line 8.

Even basic mods, such as the presented example, can be used to generate a wide range of environments. Furthermore, advanced community-made extensions and games can be easily integrated into Craftium, significantly expanding its potential. Section 3.5 highlights some of these possibilities. For detailed instructions on creating Craftium environments please refer to the online documentation (see Section 1). Finally, note that the creation of Minetest mods is outside the scope of this paper, as comprehensive resources are already available (Minetest Team, 2024a; Ward, 2023b).

## 3.3 INTERFACE

Once created, Craftium environments are used via the Gymnasium (Towers et al., 2024) (single-agent) or PettingZoo (Terry et al., 2021) (multi-agent) interfaces. Both interfaces are open-source and have become the standard interface for RL and MARL environments, providing a unified abstraction over environments that enables interoperability between environments and methods. Just by implementing these interfaces, Craftium is already compatible with many existing tools and projects to train, test, develop, and analyze many algorithms, including but not limited to `stable-baselines3` (Raffin et al., 2021), Ray RLlib (Moritz et al., 2018), CleanRL (Huang et al., 2022b), and `skrl` (Serrano-Muñoz et al., 2023).

```python
1  import gymnasium as gym
2  import craftium
3
4  env = gym.make("Craftium/Room-v0")
5
6  obs, inf = env.reset()
7  for t in range(5000):
8    a = agent(obs)
9    obs, r, tm, tc, inf = env.step(a)
10
11   if tm or tc:
12     obs, inf = env.reset()
13
14  env.close()
```

Figure 6: Python code illustrating the typical interaction loop between an agent and a Craftium environment using the Gymnasium interface.

Figure 6 illustrates an example using the Gymnasium (single-agent) interface. PettingZoo employs a very similar interface described in Appendix D. Line 4 loads an example Craftium environment by name (see Section 3.5). Line 6 initiates an episode, obtaining the first observation and a Python dictionary with additional information (e.g., elapsed time). Lines 7-12 implement the agent-environment interaction loop. In line 8, the agent selects an action based on the current observation. The line 9 executes the action specified by the agent, resulting in an observation, a reward, a truncation flag, a termination flag, and a new information dictionary, respectively.

---

[5]Map generators are documented at: `https://wiki.minetest.net/Map_generator`.

The truncation flag indicates if the maximum number of timesteps allowed by the environment is reached, while the termination flag determines if the episode has reached a terminal state (e.g., the player dies). Both flags are checked in line `11`, and if one or both of them are true, the episode is restarted in line `12`. Finally, the last line closes the environment after the main loop ends.

## 3.4 Performance

As stated in the introduction, computationally efficient environments are key for research on autonomous agents in general. Therefore, this matter has been a focal point of Craftium's development. Figure 7 compares the steps (i.e. interactions) per second obtained by Craftium to VizDoom and MineDojo, well-known environment creation platforms from the literature. Results show the average of 5 runs in 3 different environments per framework, in a machine with a single NVIDIA A5000 GPU and an Intel Xeon Silver 4309Y CPU. Craftium achieves very competitive results compared to VizDoom, even though VizDoom is based on ZDoom, which is not 3D *per se*.[6] Comparing Craftium's performance to MineDojo's, we observe that the presented framework achieves +2670 steps per second more. One of the main reasons behind this significant difference is that MineDojo relies on Minecraft, which is implemented in Java, while, Craftium is based on Minetest, implemented in C++ with low-resource machines in mind. Moreover, Minetest is open source, al-

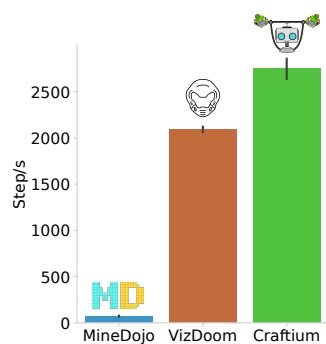

Figure 7: Average steps per second obtained with MineDojo, VizDoom, and Craftium (the greater the better).

lowing us to modify it to efficiently integrate it into our framework (see Appendix A). Contrarily, MineDojo, which internally uses MineRL (Guss et al., 2019), adds more layers of complexity to convert the original Minecraft game into an RL environment. Refer to Appendix E for more details on this analysis.

## 3.5 Illustrative examples

Much like game engines are tools for creating new games, Craftium is a general-purpose platform for developing environments. Therefore, this section highlights the potential of Craftium across various use cases: from single and multi-agent RL tasks (Sections 3.5.1 and 3.5.2 respectively) to open-world environments for large multimodal model-based agents (Section 3.5.3), and environment generators for CRL (Section 3.5.4). These examples are purely illustrative and are not presented as benchmarks. The aim is to demonstrate the framework's capabilities and provide accessible, well-documented foundations for building custom environments tailored to specific research needs.

### 3.5.1 Example 1: classic single-agent RL

This section provides examples of using Craftium to create single-agent environments for RL. We implement five tasks of diverse nature: simple environments for testing RL algorithms, sparse reward and exploration scenarios, and a challenging survival task. For simplicity, all tasks share the same $64 \times 64$ pixel RGB image observation space. Moreover, the default action space described in Section 3.1 is simplified to only use the necessary actions to solve each task, see Appendix B.2. Refer to Appendix F.1 for figures and extended descriptions of the environments.

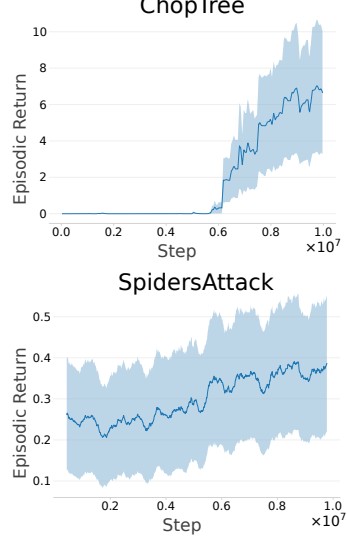

Figure 8: Episodic return curves of PPO in the *ChopTree* and *SpidersAttack* tasks. Results aggregate 5 different runs per task: average is denoted with lines and the standard error with the contour.

---

[6]See https://en.wikipedia.org/wiki/Doom_(1993_video_game).

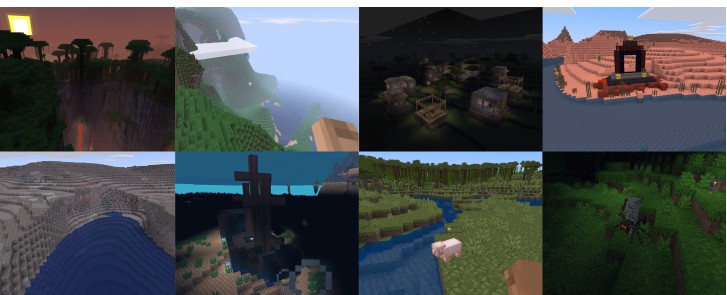

Figure 10: The leftmost picture shows an overview of the map for the open-world environment example. Each pixel in the map's picture corresponds to a single block (voxel). The map shows an area of 1.6K×1.6K blocks from the vast 64K×64K×64K blocks area that agents can explore. The color of each pixel is used to denote different biomes, some of which are visualized in the rightmost figures. The rightmost images provide a closer visualization of the included forests, cliffs, deserts, villages, underwater wreckage, monsters, etc.

To complement this example, Figure 8 demonstrates how environments of varying levels of difficulty can be designed within Craftium. The figure shows the results obtained by the Proximal Policy Optimization (PPO) algorithm (Schulman et al., 2017) in two of the presented tasks. In the *ChopTree* task, the high episodic return values indicate that PPO successfully solves the task, chopping more than 6 trees on average (a reward of +1 is given for every chopped tree). In the *SpidersAttack* scenario the agent has to survive hostile spiders. In this case, a reward value of 1 is given for every defeated spider, and 0 otherwise. As can be seen in the figure, although the episodic return value increases over time, the final average value is below 0.5. This indicates that the trained agent does not survive a single spider in half of the episodes, showing that it is a considerably more challenging task compared to the previous one. See Appendix F.1 for further details and experimental results in the rest of the tasks.

### 3.5.2 EXAMPLE 2: MULTI-AGENT REINFORCEMENT LEARNING

The previous section focuses on single-agent scenarios for RL, now, we focus on MARL, showcasing Craftium's multi-agent features. For this purpose, we develop a one vs one multi-agent combat environment in Craftium. Likewise the single-agent tasks from the previous section, the environment employs a $64 \times 64$ RGB image observation space and a simplified discrete action space. In this case, agents are rewarded (+1) when punching other agents and penalized on damage (-0.1).

To illustrate a use case, we train the agents using self-play (Crandall & Goodrich, 2005), a popular method for this type of competitive scenarios (Silver et al., 2017; 2018; Jiang et al., 2024). Results are presented in Figure 9, where the policy has been trained to play against itself using PPO. The increasing episodic return curve in the figure shows how the policy learns to fulfill the task. Refer to Appendix F.2 for figures and more details on the environment and the learning method.

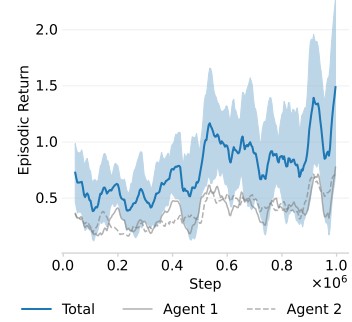

Figure 9: Average and standard error episodic return curves across 5 runs in the multi-agent combat environment. The blue line indicates the total episodic return of both agents, while the gray lines show the values separately.

### 3.5.3 EXAMPLE 3: OPEN-WORLD ENVIRONMENTS

This section introduces an open-world environment as an example of a complex scenario for embodied AI similar to MineDojo. The environment employs the open-source VoxeLibre game for Minetest, which is greatly inspired by Minecraft, sharing many similarities (Fleckenstein et al., 2024). VoxeLibre provides a rich and vast environment with many complex interactions, differ-

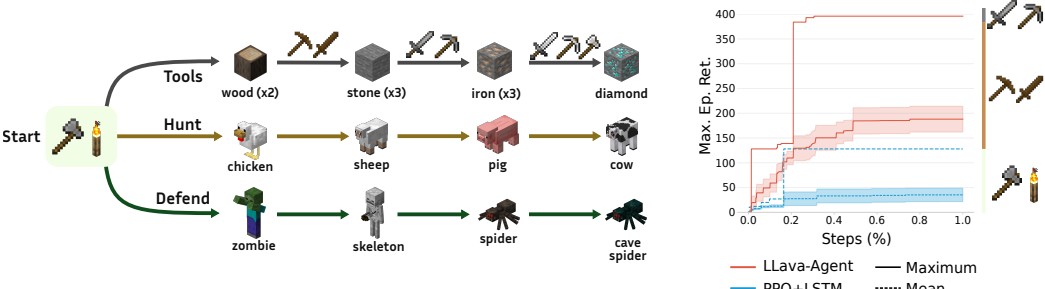

Figure 11: The leftmost diagram depicts the skills tree of the open-world environment example described in Section 3.5.3. The rightmost plot shows the results of PPO+LSTM and LLava-Agent (zero-shot) obtained in terms of average and best maximum episodic return values obtained in 10 repetitions per method. Refer to Appendix F.3 for further details.

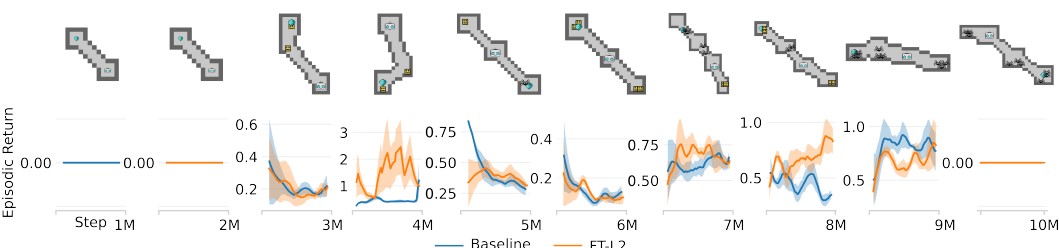

Figure 12: Episodic return curves of a baseline trained from scratch and FT-L2 over a series of 10 different environments created using the procedural generator from Section 3.5.4. Environments are illustrated above their corresponding curves (simplified 2D top-views), with the robot indicating the initial position of the agent, the diamond (the objective), and different enemies. See Appendix F.5 for details and larger visualizations of the environments.

ent biomes, animals, plants, or hostile creatures. This section also serves as an example of how community-made Minetest games can be integrated into Craftium.

The leftmost picture of Figure 10 shows an overview of the specific world map generated for this environment, where colors indicate different biomes: dark green for forest biomes, light brown for sand desert, white for artic biomes, etc. The rightmost images showcase the complexity and richness of this environment. Figure 11 presents the skills tree developed for this environment, showing the hierarchical sequence of skills that the agent can develop to reach more complex goals. Every time the agent unlocks a skill of the tool branch (e.g., collect two wood blocks) it receives a reward and new tools (e.g., wood pickaxe and sword), while the objective switches to the next skill (e.g. collect two stone blocks). Regarding the hunt and defend branches, the agent receives a reward according to the difficulty of hunting the animal or defeating the monster (refer to Appendix F.3 for details).

To complete this example, Figure 11 compares the achievements obtained by PPO (using LSTM-based memory) and an agent based on the open-source large multimodal model LLaVa (Liu et al., 2024a) version 1.6 (Liu et al., 2024b) (zero-shot: with no finetuning to this specific task). Results show that the LLaVa-Agent unlocks the collect wood and stone stages, while PPO only completes the first stage. Both methods successfully hunt animals and fight some monsters, indicated by the smaller increases in the best episodic return values in the figure. This example demonstrates Craftium's usage beyond RL, using it to analyze and evaluate the ability of large multimodal model-based agents to leverage world knowledge to approach complex open-world tasks.

### 3.5.4 EXAMPLE 4: PROCEDURAL ENVIRONMENT GENERATION FOR CRL

In CRL, agents face a sequence of environments, interacting with one at a time and limited by a timestep budget, where methods are expected to leverage prior knowledge to solve incoming tasks efficiently. Commonly employed settings rely on hand-crafted sequences, with a small number of environments, or use repetition for generating larger sequences, e.g., Wołczyk et al. (2021) and

Table 1: Popular environment frameworks compared by: number of playable dimensions, procedural generation capabilities, environment creation, whether environments can be programmatically implemented (and not through predefined configuration options), Gymnasium support, multi-agent, and open-world capabilities. We specify the language if a framework allows programmatic implementation of environments, and a red cross otherwise.

| FRAMEWORK | DIMS. | PROC. GEN. | ENV. CREAT. | PROG. DEF. | GYMNASIUM | MARL | OP. WORLD |
|---|---|---|---|---|---|---|---|
| ALE (Bellemare et al., 2013) | 2D | ✗ | ✗ | ✗ | ✔ | ✔ | ✗ |
| DM LAB (Beattie et al., 2016) | 3D | ✗ | ✔ | Lua | ✗ | ✗ | ✗ |
| AI2-THOR (Kolve et al., 2017) | 3D | ✔ | ✔ | ✗ | ✗ | ✔ | ✗ |
| VIZDOOM (Wydmuch et al., 2019) | 2.5D | ✗ | ✔ | ZScript | ✔ | ✔ | ✗ |
| MINERL (Guss et al., 2019) | 3D | ✔ | ✗ | ✗ | ✗ | ✗ | ✔ |
| NLE (Küttler et al., 2020) | 2D | ✔ | ✗ | ✗ | ✗ | ✗ | ✔ |
| PROCGEN Cobbe et al. (2020) [7] | 2D | ✔ | ✔ | ✗ | ✔ | ✗ | ✗ |
| MINIHACK (Samvelyan et al., 2021) | 2D | ✔ | ✔ | des-file format | ✗ | ✗ | ✗ |
| MINEDOJO (Fan et al., 2022) | 3D | ✔ | ✔ | ✗ | ✗ | ✗ | ✔ |
| HABITAT 3.0 (Puig et al., 2024) | 3D | ✔ | ✔ | ✗ | ✗ | ✔ | ✗ |
| CRAFTAX (Puig et al., 2024) | 2D | ✔ | ✗ | ✗ | ✗ | ✗ | ✔ |
| **CRAFTIUM** | **3D** | ✔ | ✔ | **Lua** | ✔ | ✔ | ✔ |

Tomilin et al. (2023). Consequently, this section leverages Craftium's versatility to implement a procedural environment generator that automatically produces a sequence of increasingly difficult environments for CRL (see Figure 12). The generator, given some input parameters, randomly generates labyrinthic 3D dungeons populated with hostile enemies. In these environments, the agent has to survive and reach its objective: a diamond. Every time the agent reaches the objective, a reward value of 100 is provided, of 1 when defeating an enemy, and of 0 otherwise. Further information on the generator and the environments is provided in Appendix F.4.

To complement this example, Figure 12 shows the results of an agent trained from scratch in each environment (referred to as the baseline) and an agent that finetunes the model learned in the previous task, referred to as FT-L2 (fine-tuning with L2 regularization) (Gaya et al., 2023; Wołczyk et al., 2024). As can be observed, FT-L2 greatly outperforms the baseline in some of the environments. This demonstrates how procedural processes can be implemented in Craftium to generate environment sequences for CRL that show knowledge transferability. Note that this generator can be extended beyond CRL to other scenarios such as meta-learning, open-endedness, and UED (Dennis et al., 2020; Team et al., 2021; Bauer et al., 2023; Rigter et al., 2024)

## 4 RELATED WORK

Table 1 includes a comparative overview of popular environment frameworks of the RL, open-ended, and embodied AI literature. The following lines provide a more extensive discussion of this analysis.

As stated in the introduction, a wide range of environments have been proposed for developing and evaluating autonomous agents. However, many of these environments are adaptations of video games (Bellemare et al., 2013; Wydmuch et al., 2019; Guss et al., 2019; Küttler et al., 2020) not originally designed for research. As a result, they offer limited customization, often restricted to predefined parameters (e.g., number of enemies). Examples include ALE (Machado et al., 2018), MineRL (Guss et al., 2019), and NLE (Küttler et al., 2020). The lack of flexibility hinders their use in various research scenarios, such as designing custom environments to study catastrophic forgetting or analyzing specific behaviors of open-ended learning systems.

These limitations have long been recognized, and several frameworks have been proposed that allow the creation of completely new environments. For example, VizDoom (Wydmuch et al., 2019) allows defining environments using ZScript, and MiniHack (Samvelyan et al., 2021) employs the `des-file format` for the same purpose. Both, ZScript and the `des-file format` are *Domain Specific Langauges* (DSL) tailored to the games they originate from (ZDoom and NetHack respectively). However, DSLs are often purpose-specific and lack the flexibility and functionality of general-purpose programming languages. For instance, the `des-file format` is not a program-

---

[7]Although the original project is unmaintained, the table considers the community rewrite available at https://github.com/Farama-Foundation/Procgen2.

ming language *per se*, just a language to define NetHack levels. Additionally, DSLs often differ significantly from mainstream programming languages, which limits their usability and adoption.

Some frameworks offer customization through the programming languages they are implemented in, avoiding the limitations of DSLs. For example, Griddly (Bamford et al., 2020) and MiniGrid (Chevalier-Boisvert et al., 2023) offer Python APIs for creating 2D grid-like environment. While grid environments are fast to simulate, they lack the complexity and diversity of more advanced environments like MineRL and VizDoom. Although more complex tasks could be implemented in these frameworks, it would require significant development effort for researchers. Regarding 3D environments, MiniWorld (Chevalier-Boisvert et al., 2023) offers a similar API to MiniGrid, but suffers from the same issues regarding the implementation of richer environments.

The field of embodied AI for robotics has long recognized the importance of visually complex scenarios, which include popular frameworks such as AI2-THOR (Kolve et al., 2017) and Habitat 3.0 (Puig et al., 2024). However, these works focus on accurate physical modeling and photorealism while having limited diversity (mostly including indoor household scenarios) and a lack of open-world environments. For higher-level cognitive tasks that do not require accurate physics modeling or photorealism, the field has popularly adopted Minecraft: an extremely popular game, with rich content, diverse open worlds, and complex game mechanics. Some examples are the Malmo (Johnson et al., 2016) project and MineRL (Guss et al., 2019) that wraps Minecraft in a Python interface. However, they lack support for task customization or the creation of new environments. More recently, MineDojo (Fan et al., 2022) has greatly improved customization within Minecraft-based environments. Nevertheless, environment creation is constrained by predefined parameters, making scenarios like those in Section 3.5 infeasible to implement (see Appendix G for details) and lacking multi-agent support, which hinders its adoption in this growing field.

## 5 CONCLUSION

Designing new environments and modifying existing ones is crucial for advancing research in open-ended and embodied AI, RL, MARL, and autonomous agents in general. However, many established environments provide limited or no options for customization (Bellemare et al., 2013; Johnson et al., 2016; Guss et al., 2019; Küttler et al., 2020; Matthews et al., 2024). Although some works offer tools for developing environments, they rely on restrictive DSLs (Wydmuch et al., 2019; Samvelyan et al., 2021) or simplistic 2D worlds (Bamford et al., 2020; Chevalier-Boisvert et al., 2023). Conversely, rich and complex 3D environments like MineDojo (Fan et al., 2022) allow limited customization, have no multi-agent support, and are built on closed-source and computationally expensive games like Minecraft.

This work presents Craftium, an easy-to-use and flexible framework for creating rich 3D environments. Craftium's versatility is showcased in Section 3.5, which shows its application to train and analyze single and multi-agent RL algorithms, implement open-world environments for complex embodied agent tasks, and used to procedurally generate environments for CRL. Unlike many alternatives built on top of existing video games, Craftium is based on Minetest, a fully-featured open-source game engine. This analogy is also translated to the presented framework, as it is not a benchmark but a general-purpose tool for creating environments. By leveraging the extensive and well-documented Minetest Lua API (Minetest Team, 2024a; Ward, 2023b), Craftium enables nearly limitless possibilities for the development of custom single and multi-agent environments. Additionally, Minetest has a vibrant community that has produced numerous games and extensions (Ward, 2023a), which can be easily integrated into Craftium environments (see Section 3.5.3). Moreover, its efficient implementation significantly reduces the computational cost of other alternatives. As shown in Section 3.4, Craftium achieves over 2K timesteps per second more than MineDojo, and performs competitively with VizDoom, even though VizDoom is not fully 3D. Craftium also implements the widely-adopted Gymnasium (Towers et al., 2024) and Petting Zoo (Terry et al., 2021) interfaces, making it compatible with numerous existing tools and projects, such as, Moritz et al. (2018); Huang et al. (2022b); Serrano-Muñoz et al. (2023) and Raffin et al. (2021). Finally, Craftium is open source and provides extensive documentation, including many practical examples from which users can build environments for their particular research needs.

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

## A   MODIFICATIONS TO MINETEST

Although Minetest is an extremely flexible game engine with extensibility built into its core, adapting it to be a platform for RL environment creation required modifying the game engine's source code. As Minetest is a large C++ project with thousands of files, modifications have been thoughtfully limited to as few changes as possible to ensure seamless updates to new versions of Minetest and improve maintainability. Most of the introduced code is limited to a single `craftium.h` and a few modifications to the `client.cpp` file. These modifications have allowed the implementation of features required for training RL agents in Minetest:

- Implementation of a client that connects to the Python process with the agent's implementation. This is the communication channel from which Minetest sends RGB frames and other timestep data to Python, and Python sends the next actions to be executed.

- Executing agent's actions as keyboard and mouse commands in Minetest. All actions are translated as virtual keyboard keypresses or mouse movements (for moving the camera and controlling the inventory).

- Extensions to the Minetest Lua API to incorporate vital functionalities for RL environments. Extensions include 5 new Lua functions that implement functions such as setting the episode termination flag or sending reward values.

- Minetest has a client/server architecture, where the server runs the world's logic and the client interfaces with the player (e.g., game control and rendering). However, the asynchronous nature of this architecture introduced many problems to most RL agent training scenarios, as the server could update the world many times while the client was waiting for the agent to return an action. This causes many reproducibility issues and behaviors such as monsters attacking the player while the client is waiting for the agent's response. For this purpose, Craftium introduces synchronous client/server updates. This ensures that the server waits for the client to be updated before continuing with the next update, preventing the mentioned issues.

## B  ACTION SPACE DETAILS

### B.1  DEFAULT ACTION SPACE

The default action space of Craftium environments is composed of combinations of 21 keyboard actions and mouse movements on the horizontal and vertical axes. Keyboard actions are binary values, where 1 translates to a key press and 0 if not used. Available keyboard commands are listed and described in Table 2. Note that these actions are a subset of the default keyboard controls that Minetest offers[8] and its selection is inspired by the action space of MineRL (Guss et al., 2019). Mouse movements are defined by a tuple (horizontal and vertical movements) of real values in the $[-1, 1]$ interval (see Section 3.1).

Table 2: List of available keyboard actions in Craftium environments, their corresponding key in the default Minetest controls, and their description.

| ACTION | KEY | DESCRIPTION |
| --- | --- | --- |
| Forward | W | Move the player forward. |
| Backward | S | Move the player backward. |
| Left | A | Move the player left. |
| Right | D | Move the player right. |
| Jump | Space | Jump and move up. |
| Aux 1 | E | Run faster. |
| Sneak | Shift | Sneak, move downwards. |
| Zoom | Z | Zoom in at the center of the camera. |
| Dig | Left mouse button | Puch if using a weapon or mine if using a tool. |
| Place | Right mouse button | Use the pointed object if usable, otherwise attempt to build at the pointed block. |
| Drop | Q | Drop the wielded item. |
| Inventory | I | Show/hide inventory. |
| Slot [1-9] | 0-9 | Select the item in the [0-9] position of the hotbar. |

### B.2  ACTION WRAPPERS

By default, Craftium environments have a large action space with discrete (binary) and continuous values (see Section 3). However, many tasks do not require the complete default action space and can be greatly simplified by considering only the relevant actions to solve the specific task that the environment defines. Consequently, Craftium provides tools for customizing the action space of environments by using Gymnasium Wrappers.[9] Specifically, Craftium implements two wrappers: `BinaryActionWrapper` and `DiscreteActionWrapper`.

`BinaryActionWrapper` allows selecting the subset of keyboard actions (see Table 2 for the complete list) to use in the new action space. This wrapper also simplifies the continuous mouse movement actions by discretizing them into four binary actions: move the mouse left, right, up, and down. The magnitude of these movements can be chosen by the developer. For example, this wrapper allows simplifying the default $\{0, 1\}^{21} \cup [0, 1]^2$ action space into a $\{0, 1\}^3$ space where binary values correspond to: move forward, move mouse right, and move mouse left.

`DiscreteActionWrapper` allows selecting the subset of keyboard actions and discretizes the mouse movement similarly to the previous wrapper. However, in this case, actions are not binary vectors but a single discrete value. Thus, actions can not be combined as in the case of the previous wrapper. Following the previous example, instead of simplifying the default action space into $\{0, 1\}^3$ this wrapper defines the new space as $\{0, 1, 2\}$, where 0 corresponds to move forward, 1 moves the mouse to the right, and 2 moves it to the left.

---

[8]Some controls like pausing the game or opening the chat have been excluded. For additional information visit: https://wiki.minetest.net/Controls.

[9]Refer to Gymnaium's documentation for more information: https://gymnasium.farama.org/api/wrappers/action_wrappers/.

## C  EXTENSIONS TO THE MINETEST LUA API

Minetest counts with an extensive and powerful Lua API (Minetest Team, 2024a) that can be used to modify the behavior of the game engine and create mods or entire games (Ward, 2023a). However, Minetest lacks of functionalities to define RL environments by itself. Therefore, Craftium distributes a modified version of the game engine (see Appendix A) that includes additional functionalities in the Lua API to make it possible to implement RL environments from Minetest mods. Table 3 lists and describes the new functions added to the Lua API.

Table 3: List of the new functions added to the Minetest Lua API. The "—" character is used to indicate that a function takes no arguments.

| NAME | PARAMETERS | DESCRIPTION |
| --- | --- | --- |
| set_reward | float | Sets the reward value to the given value until another call to a function that modifies the reward is made. |
| get_reward | — | Returns the reward value of the current timestep, and nil if not set. |
| set_reward_once | float, float | Sets the reward to the first parameter only for the current timestep, resetting it to the second parameter afterwards. |
| set_termination | — | Sets the termination flag to true for the current timestep. |
| get_termination | — | Returns a 1 if the termination flag is set to true, 0 otherwise. |

## D  USING CRAFTIUM THROUGH THE PETTINGZOO (MULTI-AGENT) INTERFACE

```python
from craftium import pettingzoo_env

env = pettingzoo_env.env(
    env_name="Craftium/MultiAgentCombat-v0"
)

env.reset()

for agent_id in env.agent_iter():
    observation, reward, termination, truncation, info = env.last()

    if termination or truncation:
        break

    action = agents[agent_id](observation)
    env.step(action)

env.close()
```

Figure 13: Python code illustrating an example multi-agent scenario using the PettingZoo interface in Craftium.

Figure 13 shows an example use case of the PettingZoo[10] API in Craftium for multi-agent environments. Note that PettingZoo is greatly inspired by Gymnasium and shares many similarities and design choices.[11]

Likewise the Gymnasium example from Figure 6, the first lines (1-5) instantiate a Craftium environment by name. In this case, *Craftium/MultiAgentCombat-v0* is loaded, corresponding to the multi-agent environment example showcased in Section 3.5.2. Then, line 7 resets the environment to the initial state, initializing Minetest for the first time internally. Next, lines 9-16 define the main agent-environment interaction loop. As defined in line 9 the loop cycles through the agents (two agents

---

[10]More information at: https://pettingzoo.farama.org/api/aec/.

[11]In fact, both projects are developed under the same Farama foundation, see https://farama.org/.

for this specific environment). Line 10 obtains the observation, reward, termination/truncation flags, and the information dictionary (similarly to the Gymnasium example). Next, lines 12-13 check if the episode should terminate. If the episode continues, line 15 selects the action for the current agent, and line 16 executes it, running a single environment step for the current agent. Finally, 18 closes the environment, shutting down Minetest and removing any temporal files.

# E   DETAILS ON THE PERFORMANCE BENCHMARK

Due to the page limit constraint of the paper, this section provides additional details on the environment performance comparison presented in Section 3.4.

To complement the results illustrated in Figure 7, Table 4 provides the exact average and standard deviation values. The measurements aggregate the results of 5 different runs of 1K steps in 3 environments per framework. Note that all environments considered for this experiment were single-agent, as MineDojo does not support multi-agent scenarios[12] and VizDoom does not provide multi-agent environments (although technically supports this setting).[13] The environments were: speleo, room, and spiders attack for Craftium (see Appendix F.1); *VizdoomHealthGathering-v0*, *VizdoomCorridor-v0*, and *VizdoomDefendCenter-v0* for VizDoom; and *harvest_milk*, *creative:255*, and *Harvest* for MineDojo. In all cases, observations were RGB images, without frameskip, and actions were selected uniformly at random. In the case of MineDojo and Craftium environments observation size was set to $64 \times 64$ pixels, and to $320 \times 240$, as the latter resolution is not available for VizDoom environments.

As can be observed in Table 4, Craftium achieves substantially higher, +38%, steps per second than the Minecraft alternative, MineDojo. The reasons for such a significant performance gap are many, as both frameworks are complicated systems with many interacting components. One of the most significant differences is the choice of implementation language: MineDojo is based on Minecraft, which is implemented in Java 8, while Craftium relies on Minetest, implemented in C++ and known to perform significantly higher than Java.[14] Another relevant aspect is that Minecraft is a *complete* game, that has grown in complexity over the years, and this complexity directly affects to the environments implemented on top of it. As it is a close source game, the developer is not allowed to modify its source code to remove irrelevant parts of the game for the environment at hand for the sake of computational efficiency. Contrarily, Minetest is open source and exposes a highly flexible Lua API to modify its behavior. This allows building environments with only the relevenat components for the task at hand. Along the same line, the open-source nature of Minetest allowed its modification to tightly integrate it with the proposed framework. For example, to incorporate a system to execute the actions sent from the Python interface as keyboard and mouse commands. Conversely, Minecraft does not allow modifications to its source code, which requires MineRL and MineDojo[15] to include many layers of complexity to adapt the Minecraft game to the RL setting. Most notably, Minecraft is a game and is not intended to run on a server without a monitor. Therefore, MineRL and MineDojo use an external tool, *Xvfb*[16]. to emulate a monitor without showing any screen output, which causes significant performance drawbacks. This also implies that the X11 windowing system[17] is installed, which is not often the case in HPC clusters.

# F   DETAILS ON THE ILLUSTRATIVE EXAMPLES

Due to the size limitations of the main paper, this section includes additional information on the illustrative examples shown in Section 3.5.

---

[12]Revelant discussion at (accessed November 2024): `https://github.com/MineDojo/MineDojo/issues/15`.

[13]For more details on the multi-agent capabilities of VizDoom (accessed November 2024): `https://github.com/Farama-Foundation/ViZDoom/issues/546`.

[14]For example, see the performance comparison at `https://benchmarksgame-team.pages.debian.net/benchmarksgame/fastest/gpp-java.html`.

[15]Note that MineDojo is based on MineRL. Refer to the work by Fan et al. (2022) for details.

[16]See `https://en.wikipedia.org/wiki/Xvfb`.

[17]See `https://en.wikipedia.org/wiki/X_Window_System_core_protocol`.

Table 4: Average and standard deviation values obtained in the environment framework performance comparison conducted in Section 3.4.

| FRAMEWORK | STEP/S |
|---|---|
| CRAFTIUM | **2746.69±230.41** |
| VIZDOOM | 2091.91±59.03 |
| MINEDOJO | 71.87±11.82 |

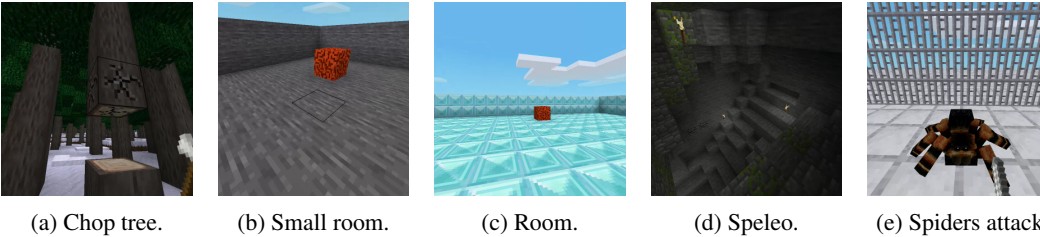

(a) Chop tree.     (b) Small room.     (c) Room.     (d) Speleo.     (e) Spiders attack.

Figure 14: Visualizations of the example environments for classic single-agent RL.

### F.1 ENVIRONMENTS FOR CLASSIC SINGLE-AGENT RL

All tasks share the same observation space of $64 \times 64$ pixel RGB images. In all cases, the action space has been simplified into a discrete space $a \in \{0, 1, 2, \ldots\}$ as described in Section 3.1 (see Appendix B.1 for details). The simplified action space also introduces a nop action (do nothing) to all tasks. The following lines describe the five tasks introduced in this section.

**Chop tree.** The agent is placed in a dense forest, equipped with a steel axe (see Figure 14a). Every time the agent chops a tree, a positive reward of +1 is given, 0 otherwise. Therefore, the task is to chop as many trees as possible until episode termination. Available actions are nop, move forward, jump, dig (used to chop), and move the mouse left, right, up, and down. Episodes terminate when 2K timesteps are reached.

**Room and small room.** These tasks present the same objective in different scenarios. In both cases, the agent is placed in one half of a closed room with a red block in the other half of the room. The objective is to reach this block as fast as possible. The difference between both tasks is the size of the room (see Figures 14c and 14b). The reward is constant, all timesteps have a reward value of -1, and the episode terminates when the agent reaches the block. To avoid solving the task by memorization, the initial position of the agent and the red block are randomized in every new episode. Available actions are: move forward, move mouse left, and move mouse right. The timestep budget is 1K in *SmallRoom*, and 2K for the variant with the larger room. Four actions are available: nop, move forward, and move the mouse right and left.

**Speleo.** The agent is located in a closed cave illuminated with torches (see Figure 14d). The task is to reach the bottom of the cave as fast as possible. For this purpose, the reward at each timestep is the negative altitude (Y-axis position) of the agent. Therefore, the reward increases as the agent goes deeper into the cave. Actions are nop, move forward, jump, and move the mouse left, right, up, and down. Episodes terminate if the agent dies (falling from a great height) or if 3K timesteps are reached.

**Spiders attack.** The agent is placed in a large cage together with hostile spiders (see Figure 14e), it is equipped with a steel sword and the objective is to survive. In the beginning, there is a single spider in the cage, but every time all spiders are defeated, a new round starts with one more spider than in the previous one (until 5 spiders). The reward of defeating a siper is +1. Actions are: nop, move forward, move left, move right, jump, attack, and move mouse left, right, up, and down. Finally, episodes terminate if the agent dies or if the 4K timestep limit is reached.

Table 5: Episodic return values obtained by PPO compared to a random agent across the environments. Results show the average and standard deviation values of 5 random seeds.

| ENV. | PPO | RND |
|---|---|---|
| *Chop Tree* | $2.08 \pm 2.6$ | $0.00 \pm 0.00$ |
| *Room* | $-459.42 \pm 3.33$ | $-495.62 \pm 46.73$ |
| *Small Room* | $-180.37 \pm 16.53$ | $-250.00 \pm 0.00$ |
| *Speleo* | $-4160.22 \pm 0.26$ | $-4498.97 \pm 10.67$ |
| *Spiders Attack* | $-0.30 \pm 0.05$ | $0.00 \pm 0.00$ |

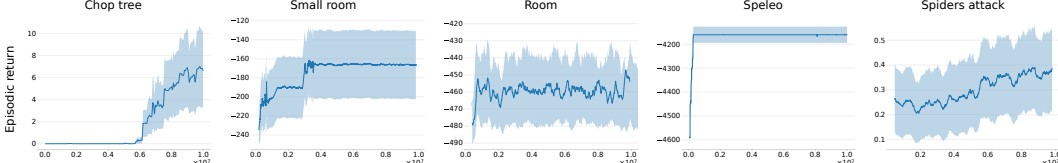

Figure 15: Episodic return curves obtained by PPO in all of the tasks from Section 3.5. Lines aggregate the average values of 5 different seeds per task, while the contour denotes the standard error of the results.

Complementing the examples from Section 3.5.1, Figure 15 provides the episodic return curves of PPO in all of the presented tasks, while Table 5 compares these results with a randomly acting agent. In both cases, results aggregate 5 runs per task, where PPO was trained for 10M timesteps in each. These experiments are mere examples to complement Section 3.5.1, and thus, no hyperparameter tuning was performed to improve the obtained results. Moreover, the performance in some of the tasks might be substantially improved if more training timesteps are considered.

Regarding the PPO algorithm, we employed the high-quality implementations from CleanRL Huang et al. (2022b). Specifically, the PPO implementation for Atari environments was adapted to Craftium environments, as both observation spaces consist of RGB images and action spaces are discrete (in the case of the environments presented in Section 3.5.1). Moreover, this implementation already considers many details shown to benefit PPO (Huang et al., 2022a). The hyperparameters and CNN network architecture were set according to their default values in the original PPO implementation from CleanRL.[18]

### F.2   MULTI-AGENT COMBAT

This section describes the multi-agent environment example from Section 3.5.2 in detail. As can be seen in Figure 16, the scenario consists of a completely flat world, where two agents are placed in a closed jail. Both agents have no items or tools available, and cannot escape the jail. Similarly to the classic single-agent RL task (see Section 3.5.1 and Appendix F.1), observations are $64 \times 64$ RGB images, and the action space consists of a simplified discrete space using the `DiscreteActionWrapper` from Appendix B.2. Specifically, the discrete action space consists of the following actions: nop, forward, left, right, jump, attack, and move the mouse right or left. An agent gets a positive reward (+1) when punching other agents and (-0.1) on damage (i.e., losing one health point). Finally, episodes terminate if the number of health points (initialized to 20) of any of the agents is zero, or the maximum number of timesteps (2K by default) is reached.

Regarding the self-play method employed in Figure 9, we employ exactly the same CNN architecture and PPO algorithm implementation as in the single-agent environment examples from Appendix F.1 (refer to the last part of this appendix for details). In this case, as we employ self-play (Silver et al., 2017), both agents share the same internal NN-based policy, which is updated every 128 steps. Finally, the agents were trained for 1M timesteps using grayscale versions of the observations and frame staking of 4 frames, resulting in a $4 \times 64 \times 64$ pixel observation space.

---

[18]Souce code of the original PPO implementation: `https://github.com/vwxyzjn/cleanrl/blob/38c313f8326b5049fe941a873e798485bccf18e5/cleanrl/ppo_atari.py`.

Figure 16: Screenshot of the illustrative multi-agent environment from Section 3.5.2.

### F.3 OPEN WORLD

In Section 3.5.3 we introduce an open-world environment. In this environment, the agent has to survive and gather resources in an open world based on the open-source VoxeLibre (Fleckenstein et al., 2024) game for Minetest. The environment is designed to have three different tracks: tools, hunt, and defend.

The first, the *Tools* track, consists of 4 different milestones: collect two wood blocks, three stone blocks, three iron blocks, and finally, a diamond block. When the agent unlocks one of the stages (i.e. tasks) it receives a reward and a new set of tools to employ to solve the next task. The reward for completing each of the stages is 128, 256, 1024, and 2048 respectively. Moreover, when the agent unlocks a new stage, it receives a sword and a pickaxe of the material of the completed stage. For example, if the agent unlocks the *wood* stage (collect two wood blocks), a wood sword and pickaxe are automatically added to its inventory. To simplify solving the first stage of this track, the initial inventory of the agent is composed of a stone axe and 256 torches. The stone axe allows the agent to more easily chop trees to collect wood, while it also serves to defend from enemies (i.e. monsters) and hunt animals.

Conversely, the *Hunt* and *Defend* tracks are non-sequential. The agent is expected to develop skills to handle increasingly complex scenarios rather than progressing linearly (although this could also be the case). In these tracks, a reward is provided to the agent every time it *puches* an enemy or an animal. In the case of enemies, the reward value is equal to the damage caused by the tool, while in the case of the animals, this value is reduced to half. The motivation behind this particular reward function is the following. If the agent defeats an enemy or hunts an animal, the episodic return obtained by the agent is linear to the life of the enemy or animal. Moreover, the agent is also encouraged to use the correct tool for these tasks. For example, using a sword to fight a monster will provide more reward than using a torch or pickaxe for the same task.

In Minetest, the time of day of the game is linked to the real clock time, where the day/night cycle lasts for 20 minutes by default.[19] In consequence, in this environment the time of day is set according to the global timestep to maintain consistency and avoid relaying in real clock time while training agents. If the latter is not considered, the time of day experienced by the agents could vary depending on the time required by the agent to select an action, which greatly varies depending on its implementation and architecture.

The following lines provide details on the methods used in the experiment from Section 3.5.3. Note that in both cases, the action space of the agents was composed of 18 discrete actions, defined using `DiscreteActionWrapper` from Appendix B.2. The actions are: nop, move forward, backward, left, and right, jump, sneak, dig, place, slot 1, slot 2, slot 3, slot 4, slot 5, move the mouse

---

[19]Additional information at `https://wiki.minetest.net/Time_of_day`.

right, left, up, and down. Slot $[1, \ldots, 5]$ corresponds to the actions of selecting the tool or object in that position of the inventory (i.e. often referred to as the *hotbar*).

**PPO+LSTM.** This method is based on the popular PPO algorithm while employing a convolutional neural network to encode observations and an LSTM module providing memory capabilities to the agent. As the experiments described in Appendix F.1, this agent is based on CleanRL's PPO implementations, in this case in PPO+LSTM for Atari games.[20] Similarly, hyperparameters were kept fixed (not optimized), as the purpose of this experiment is to serve as example. Finally, the observation space for this agent was set to $84 \times 84$ of greyscale images using 4 observations for frame stacking.

**LLaVa-Agent.** This agent is based on the open-source large multimodal model (LMM) LLaVa by Liu et al. (2024a), specifically version 1.6 (Liu et al., 2024b). This agent is not intended as a new proposal for LMM for embodied AI, but just as an example of how LMMs can be employed within Craftium environments to solve general tasks by leveraging their world knowledge. For this purpose, LLaVa has been directly employed with no fine-tuning for the open-world environment. Specifically, at each timestep, LLaVa is provided with the current observation ($512 \times 512$ pixel RGB image) and a short prompt describing the current task. The prompt also includes a list of all available actions, where LLaVa is asked to choose one. Then, the action that the agent is going to take is selected by parsing the result of the model. A random action is chosen if a parsing error occurs, although we observed that this barely happens. The employed prompt is:

> You are a reinforcement learning agent in the Minecraft game. You will be presented with the current observation, and you have to select the next action with the ultimate objective to fulfill your goal. In this case, the goal `<objective>`. You should fight monsters and hunt animals just as a secondary objective and survival. Available actions are: do nothing, move forward, move backward, move left, move right, jump, sneak, use the tool, place, select hotbar slot 1, select hotbar slot 2, select hotbar slot 3, select hotbar slot 4, select hotbar slot 5, move camera right, move camera left, move camera up, move camera down. From now on, your responses must only contain the name of the action you will take, nothing else.

Note the `<objective>` placeholder, this is replaced with the text corresponding to the current objective: "is to chop a tree", "is to collect stone", "is to collect iron", or "is to find diamond blocks". This text is automatically placed every time the agent unlocks a stage of the *Tools* branch of the skills tree.

**Details of Figure 11.** The figure aggregate results from 10 different random seeds for the PPO+LSTM method, and from 10 runs LLaVa-Agent, where each of the latter runs was constrained by a 1-hour limit ($\approx 7000$ prompting iterations per run). Consequently, the X-axis has been set to the training time percentage to accommodate both cases and for the sake of proper visualization. Finally, the Y-axis shows the best and average maximum episodic return value obtained for each method in each (normalized) training step. This choice is motivated to properly visualize when a method unlocks one of the milestones from the skills tree.

### F.4 PROCEDURAL ENVIRONMENT GENERATION

The procedural environment generation example employs a random dungeon generator implemented for this work. Although the generator can randomly create a vast number of different environments, their ultimate goal is the same. In these environments, the agent is randomly placed (equipped with a sword) in a room and has to navigate a labyrinthic dungeon full of hostile enemies (monsters) to reach the diamond. This process is divided into two steps: ① randomly generate the dungeon's map, represented in ASCII (defined in Appendix F.4.1), and ② build the 3D environment from the map.

---

[20]The original implementation can be found at: `https://github.com/vwxyzjn/cleanrl/blob/38c313f8326b5049fe941a873e798485bccf18e5/cleanrl/ppo_atari_lstm.py`.

① This first step is accomplished by the `RandomMapGen` Python class, which implements the dungeon generation algorithm. Given some input parameters, `RandomMapGen` returns an ASCII representation of the generated map. Internally, `RandomMapGen` first creates the rooms, places the enemies, and locates the objective and the agent's initial position (the agent and the objective are never located in the same room). Then, an iterative algorithm based on repelling forces is used to place the rooms such that none of them intersects with any other. Secondly, it computes the minimum number of corridors needed to create a map where all rooms are reachable. Finally, it rasterizes the map into its ASCII representation using Bresenham's line algorithm.[21]

The complete list of parameters that `RandomMapGen` accepts is the following:

- Number of rooms of the dungeon.
- Minimum and maximum sizes of the rooms. The final size is randomly selected between this range.
- A dispersion parameter in the $[0, 1]$ range that controls the distance between the rooms.
- Minimum and maximum number of monsters per room. If the minimum is set equal to the maximum, the number of monsters per room is fixed.
- The probability of each monster type of being located in one room. `RandomMapGen` considers up to 4 types of different monsters. Monster types are denoted as: `a`, `b`, `c`, or `d`. The specific monster that will be considered for each type is defined by the user in step ②.
- A boolean flag indicating whether monsters can appear in the room selected for the agent's initial position.

② Once the ASCII map is created, a mod is used to generate the final 3D dungeon inside Minetest. This mod iterates over the characters that compose the map and places the blocks and enemies (referred to as *mobs* in Minetest and gaming terminology, not to be confused with mods) accordingly. The configuration parameters of the mod are the following:

- The ASCII map generated in step ① (or via another process).
- Names of the monsters for types `a`, `b`, `c`, or `d`. Available monsters are described in the documentation of the `mobs_monsters` project.[22]
- The material used for the construction of the dungeons.[23]
- The name of the object to use as the objective (a diamond by default).[24]
- The reward of reaching the objective (100 by default).
- The reward of defeating a single monster (1 by default).

### F.4.1 THE ASCII MAP FORMAT

The ASCII map format has been intentionally designed to be human-readable and to facilitate the implementation of custom procedures to create them (or even specified by hand). The format consists of 9 possible characters, listed and described in Table 6. As can be seen in Figure 17a, maps are divided into layers, divided by the "−" (dash) character. The first layer is commonly employed to define the floor of the dungeons, while the second defines the walls and the positions of all characters and the objective, the rest of the layers are used for determining the height of the walls.

### F.5 ENVIRONMENT SEQUENCE FOR CONTINUAL RL

In Section 3.5.4, the procedural environment generation is applied to CRL by defining a sequence of related and increasingly difficult scenarios. Similarly to the examples from Section 3.5.1, the baseline and FT-L2 methods are based on the PPO implementations from CleanRL, specifically the

---

[21]See `https://en.wikipedia.org/wiki/Bresenham%27s_line_algorithm`.

[22]Accesible at: `https://codeberg.org/tenplus1/mobs_monster`.

[23]List of some available materials: `https://wiki.minetest.net/Games/Minetest_Game/Nodes`.

[24]List of some available items: `https://wiki.minetest.net/Games/Minetest_Game/Items`.

Table 6: List of characters that comprise the ASCII map format and their meaning.

| Character | Meaning |
|---|---|
| (whitespace) | Air block. |
| # | Construction block. Used for the floor and walls. |
| @ | The initial position of the agent. |
| O | Position of the objective. |
| a, b, c, d | Location of a monster of type a, b, c, or d |
| - | New layer. |

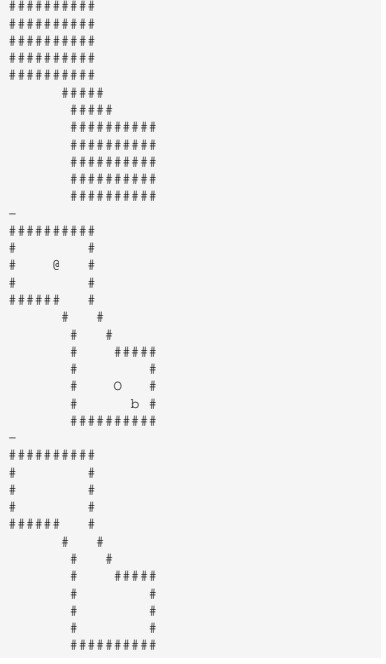

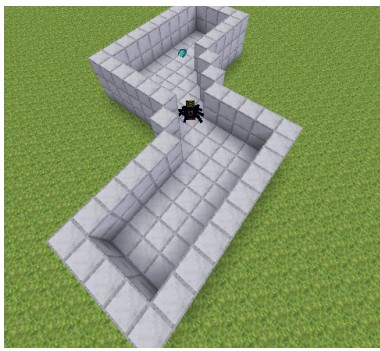

(a) ASCII map representation.  (b) Resulting 3D dungeon environment.

Figure 17: Example ASCII map format of a dungeon environment and the resulting 3D scenario in the Craftium environment. Note the 3D characterizations of the spider (denoted with a in the ASCII map) and the diamond (O in the ASCII map).

Atari ones. Where the difference between the baseline and FT-L2, is that the latter fine-tunes the model learned in the previous task and uses L2 regularization during training, while the baseline always learns a model from scratch. FT-L2 was selected for this example as it has shown significant forward knowledge transfer capabilities in other works (Gaya et al., 2023; Wołczyk et al., 2024; Malagon et al., 2024). As can be seen in Figure 12, FT-L2 substantially improves the results of the baseline in the 4th, 7th, and 8th environments, showing considerable forward knowledge transfer between different environments.

Figure 18 provides a larger 2D visualization of the environments, not included in the main paper for page limit constraints. Observing this figure we see that the first two environments employ the same map. This is intended, as the training time in each environment is low (1M timesteps), thus the first two environments offer CRL methods a way to learn to reach their objective before more difficult tasks arrive.

Regarding the observation and action spaces, they have been kept constant across the sequence. The observation space is set to 64×64 pixel greyscale images, with 4 frames for frame stacking, and the same quantity for frame skipping (Huang et al., 2022a). The action space consists of a set of 10

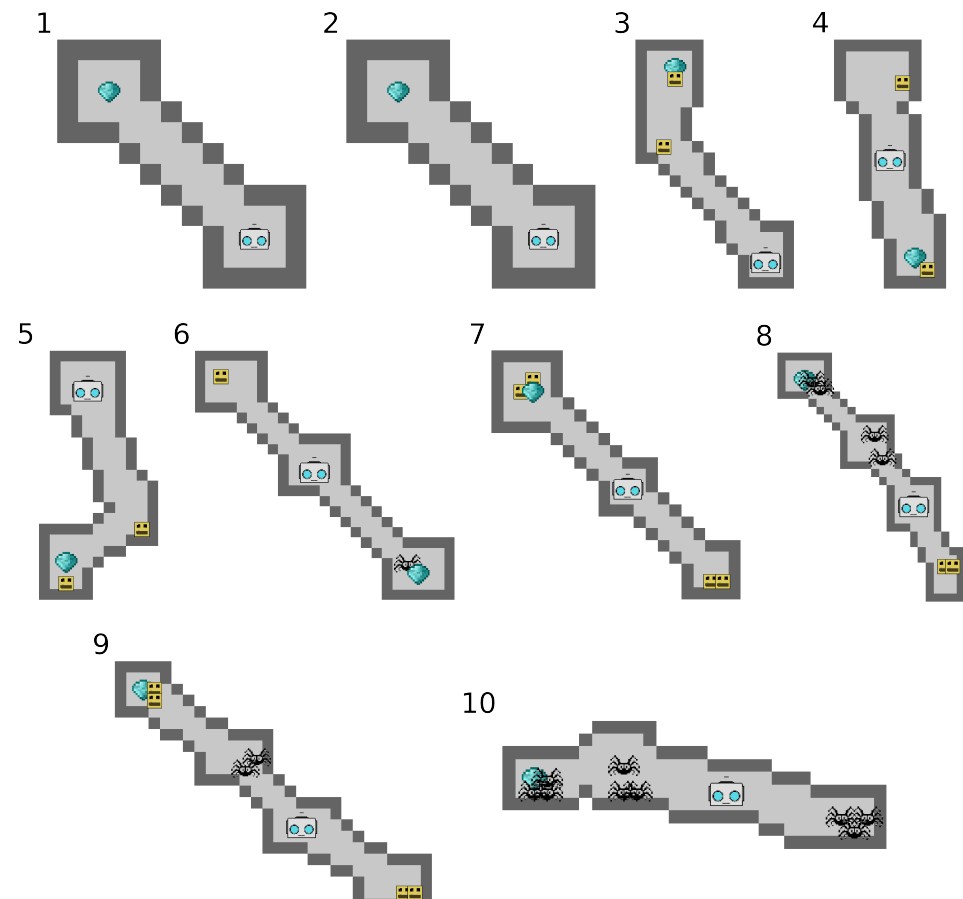

Figure 18: Overview of all the maps generated for the CRL environment sequence in Section 3.5.4. Note that these are 2D representations of the environments and that the actual environments are 3D (as can be seen in Figure 17b). The robot indicates the initial position of the agent, while the yellow characters indicate sand monsters, and the black characters denote spiders. Maps have been enumerated with their corresponding position in the CRL sequence.

discrete actions: nop, move forward, left, and right, jump, attack, move the mouse right, left, and down. Finally, episodes terminate if the health of the agent is exhausted or 5K timesteps are reached.

## G  DETAILS ON THE ENVIRONMENT CREATION CAPABILITIES OF CRAFTIUM AND MINECRAFT-BASED FRAMEWORKS

Craftium not only significantly outperforms the computational efficiency of Minecraft-based frameworks (as demonstrated in Section 3.4), but also provides an extremely flexible interface for creating new environments via the Minetest Lua API. The flexibility and versatility of this API is demonstrated by the rich and complex environments that can be created with it, see Figure 11, thanks to the wide range of mods created by the community (see Ward (2023a) for examples). This section focuses on showcasing some code examples that directly compare the flexibility of Craftium's Minetest Lua API with the MineDojo API to create new environments. Note that we only compare Craftium to MineDojo as it is, currently, the only Minecraft-based framework that allows the creating of custom environments.

One major limitation of the MineDojo's API is that although it allows for spawning different Minecraft entities (mobs and items) in a given location, the behavior, aspect, and other properties of the entities are those of Minecraft (the default ones), and cannot be changed. Figure 19 shows how MineDojo allows spawning entities. On the other hand, Craftium leverages the Minetest Lua API, which allows access to the internal state of the game engine, allowing to change any aspect of it in real time. This is illustrated with an example code in Figure 21 and Figure 22 that show how many properties and behaviors of entities can be modified in Craftium.

Another crucial difference between Craftium's and MineDojo's APIs is the map generation capabilities. MineDojo limits map generation to some predefined scenarios (only 5) and biomes. Figure 20 shows the map customization capabilities of MineDojo. On the other hand, Craftium, via the Minetest Lua API, allows the user to define any type of custom biome, and combine them in any way.[25] In Figure 23 we showcase a simple example of defining a custom desert biome in Craftium using the Minetest Lua API. Note that Craftium users can employ any of the vast number of biomes already implemented by the community (some of them illustrated in Figure 11).[26]

```
1 env.spawn_mobs("spider", [5, 0, 5])
```

Figure 19: **MineDojo.** Although MineDojo allows for spawning entities in some position, lacks the capability for modifying the behavior of entities in any way.

```
1 env = minedojo.make("open-ended", specified_biome="desert")
```

Figure 20: **MineDojo.** MineDojo only allows defining worlds from a set of predefined biomes and scenarios.

```
1 local mob_def = minetest.registered_entities["mobs_monster:zombie"]
2 mob_def.on_punch = function(self, hitter)
3     hitter:set_hp(hitter:get_hp() + 5)
4 end
```

Figure 21: **Craftium.** Example code demonstrating how the behavior of entities can be modified in Craftium. In this case, the definition of zombies is changed to increase the health of the agent by 5 when successfully attacking a zombie.

---

[25]More information and tutorials at `https://rubenwardy.com/minetest_modding_book/en/advmap/biomesdeco.html`.

[26]Examples at `https://content.luanti.org/packages/?tag=mapgen`.

```lua
1  mobs:register_mob("craftium:my_spider", {
2      docile_by_day = false,
3      group_attack = true,
4      type = "monster",
5      passive = false,
6      attack_type = "dogfight",
7      reach = 2,
8      damage = 3,
9      hp_min = 25,
10     hp_max = 25,
11     armor = 200,
12     walk_velocity = 3,
13     run_velocity = 6,
14     jump = false,
15     on_die = function(self, pos)
16         -- Set reward to 1.0 for a single timestep then reset to 0.0
17         set_reward_once(1.0, 0.0)
18         -- Spawn more spiders
19         num_spiders = num_spiders + 1
20         for i=1,num_spiders do
21             spawn_monster({ x = 3.7 - i, y = 4.5, z = 0.0 })
22         end
23     end
24 })
25
26 local monster = mobs:add_mob(pos, {
27     name = "craftium:my_spider",
28     ignore_count = true,
29 })
```

Figure 22: **Craftium.** Example of a completely custom spider type. Note that we only show a few options of those available: group attack capabilities, health, reach, attach type, armor, velocity, etc. Moreover a custom behavior is defined to set the reward and spawn more spiders when the spider dies.

```lua
1  -- Register a custom biome (e.g., desert)
2  minetest.register_biome({
3      name = "custom_desert",
4      node_top = "default:sand",
5      depth_top = 1,
6      node_filler = "default:stone",
7  })
8
9  -- Generate a random landscape with different biomes
10 minetest.register_on_generated(function(minp, maxp, blockseed)
11     if math.random() > 0.5 then
12         minetest.set_biome_area(minp, maxp, "custom_desert")
13     end
14 end)
```

Figure 23: **Craftium.** Example showing how custom biomes can be created and used in Craftium.