# OpenReview forum: "Craftium: Creating Efficient Environments for Open-Ended and Embodied Agents Beyond Gridworlds"
_ICLR.cc/2025/Conference — Submitted to ICLR 2025_

### Official Review · Reviewer_TGC5 · 2024-10-22

**Soundness:** 3
**Presentation:** 3
**Contribution:** 2
**Rating:** 6
**Confidence:** 2

**Summary:**

This paper proposes `Craftium`, a framework based on MineTest. It is similar with MineCraft but open source and efficient.

**Strengths:**

1. The simulation is efficient.
2. The writing is good and experiments is detailed.
3. This framework is easy to use.

**Weaknesses:**

1. The difference between `Craftium` and Minecraft is not highlighted in figure 2 and I think it is important. What parts of figure 2 are not supported in MineCraft?
2. The motivation of experiments are not clear.  In 3.5.2, the line 418 said
> This example demonstrates how
Craftium environments can be used to analyze and evaluate the ability of large multimodal model based agents to leverage world knowledge to approach complex open-world tasks."

Is MineCraft also able to do this thing? If it is, I don't think it is your contribution. What's the meaning of rightmost icons of Figure 11?

3. In table 1, one of the advantage of `Craftium`  is "GYMNASIUM".  Can you give a detailed analysis of how GYMNASIUM implementation helps RL training?

4. There is no detailed or systematic analysis or examples on advantages of `Craftium` over MineCraft for RL. During RL, what information can `Craftium` give for a better learning but MineCraft cannot? I think detailed internal state/information should be the advantage.

**Questions:**

- See "Weakness".
- What's the motivation of compare LLava-Agent and PPO+LSTM in Sec 3.5.2?
- Can MineCraft support RL or environment generation?
- I don't understand what's the disadvantages of MineCraft except the efficiency after reading. Can you compare the provided APIs between MineCraft and `Craftium`?

---

> ### Author Response · Authors · 2024-11-21
>
> First, we are grateful to the reviewer for the time reviewing our paper and providing the feedback. In the following lines, we address the weaknesses and questions raised by the reviewer.
>
> **[Weaknesses]**
>
> > The difference between Craftium and Minecraft is not highlighted in figure 2 and I think it is important. What parts of figure 2 are not supported in MineCraft?
>
> Based on common concerns raised by the reviewers we are happy to announce that **we have added support for multi-agent scenarios** to Craftium (refer to the general response above for details). Considering that **all Minecraft-based frameworks are single-agent**, Craftium now also differentiates itself by supporting multi-agent scenarios. Therefore, we will update Figure 2 with [this](https://drive.google.com/file/d/1Q6YICabmiXheNm0HwJh0sWcmFseIW9Va/view?usp=sharing) figure, which depicts several parts of the diagram as repeated by N, where N is the number of agents, differentiating itself from all Minecraft-based frameworks that only support a single agent. Note that Craftium provides additional features beyond multi-agent when compared to Minecraft-based alternatives, which are described in the answers below (last weakness).
>
> > The motivation of experiments are not clear. In 3.5.2, [...] Is MineCraft also able to do this thing? If it is, I don't think it is your contribution. What's the meaning of rightmost icons of Figure 11?
>
> Before Section 3.5.2, the paper focuses on the capabilities that differentiate Craftium from Minecraft-based environments (discussed in the answer to the last weakness here). In Section 3.5.2 our objective is to demonstrate that as in the Minecraft-based frameworks, Craftium can also be used to analyze the ability of LMMs to leverage general world knowledge to solve complex open-world tasks. In other words, after showcasing Craftium's many features that Minecraft environments lack, our objective in Section 3.5.2 is to demonstrate that Craftium also enables the research that can be done in Minecraft [1,2], but with all the additional capabilities that Craftium contributes.
>
> > [...] Can you give a detailed analysis of how GYMNASIUM implementation helps RL training?
>
> Note that [Gymnasium](https://gymnasium.farama.org/) is an API, an abstraction layer that allows RL practitioners to use the same code and agents for different types of environments. Before Gymnasium, the code developed for an environment (e.g., the implementation of an RL algorithm) could not be used in another environment, as environments relied on different dedicated APIs not compatible between them. With Gymnasium, practitioners can develop algorithms, tools, and libraries that work for all environments that implement the Gymnasium interface (as is the case for Craftium). These features have led to widespread usage of Gymnasium in the RL field, making it the *de facto* API for RL environments [3].
>
> > There is no detailed or systematic analysis or examples on advantages of Craftium over MineCraft for RL. During RL, what information can Craftium give for a better learning but MineCraft cannot? [...]
>
> We want to emphasize that Craftium is **substantially different** from environment frameworks based on Minecraft. Craftium provides the environment richness of Minecraft (e.g., many biomes, materials, animals, monsters, items, complex interactions, weather system, etc.) but includes **crucial** features that Minecraft-based frameworks lack:
>
> - **Efficiency.** The gap between Craftium's computational cost and Micraft-based alternatives is substantial (see Figure 7 in the paper). Importantly, this feature **should not be trivialized**, as computational resources are one of the major limitations for most AI research labs. Craftium's reduced computational cost helps to democratize access to technology and research capabilities. Computational cost is often the key to running some experiments or not. Having an extremely computationally efficient alternative to Minecraft-based frameworks, makes Craftium very relevant for many use cases, e.g., executing tens of repetitions of an experiment and large task sequences in lifelong RL. Therefore, part of Craftiium's novelty consists of providing the environment richness of Minecraft at an extremely reduced cost.

---

> > ### Author Response · Authors · 2024-11-21
> >
> > - **Minecraft is a game, and Minetest is a game engine.** Craftium is based on the Minetest game engine, while Minecraft is a game. This is a very important difference that has a profound impact on the capabilities of Craftium compared to alternatives based on Minecraft (e.g., MineDojo and MineRL). Minecraft is always the same game, in contrast, Minetest is a game engine in which we **can create completely new environments** for RL, open-ended, and embodied AI research. This directly affects the versatility of the framework: RL practitioners can use Craftium to create completely custom environments, based on existing ones or develop completely new ones. Minecraft-based alternatives do not support new environment creation (MineRL), or provide much more limited tools for this purpose (MineDojo). No Minecraft-based alternative offers Craftium's flexibility and versatility for creating environments for RL and autonomous agent research, which we strongly believe is a significant novelty.
> >
> > - **Multi-agent support.** As mentioned, we have added multi-agent support to Craftium. Note that **none of the Minecraft-based environment frameworks offers this feature**. Moreover, considering other environment creation frameworks, Craftium is the only one that combines multi-agent support, procedural generation, and such a level of environment richness, significantly enhancing Craftium's contribution and novelty. Refer to the general response above for additional information on this topic.
> >
> > Finally, in light of the reviewer's feedback, we will update the paper to clarify the differences between Craftium compared to Minecraft-based environments and the benefits of supporting Gymnasium.
> >
> > **[Questions]**
> >
> > > What's the motivation of compare LLava-Agent and PPO+LSTM in Sec 3.5.2?
> >
> > As mentioned in the answer to the second weakness, the objective of Section 3.5.2 is to demonstrate that Craftium also allows the type of research and experiments that Minecraft-based alternatives offer. For this purpose, we compare LLava-Agent and PPO+LSTM to show how the general world knowledge of LMMs can be used to solve complex open-world tasks, as these types of experiments are typically conducted in Minecraft-based environments [1,2].
> >
> > > Can MineCraft support RL or environment generation?
> >
> > It is important to note that Minecraft is a game, intended for human players, and does not support any type of integration with RL or autonomous agents by itself. Therefore, several projects have adapted Minecraft for this purpose, notably: MineRL and MineDojo. MineRL is the most popular of these, however, it only offers a set of five predefined environments with no support for creating new ones. MineDojo improved the support for generating new environments, however, this capability is limited to changing some predefined configuration options. Craftium supports creating completely new environments, **a feature that none of the mentioned alternatives has**, with an unprecedented level of flexibility thanks to the versatility of the Minetest game engine. Refer to the answer to the last weakness for more details on the differences between Craftium and Minecraft-based alternatives.
> >
> > > I don't understand what's the disadvantages of MineCraft except the efficiency after reading. Can you compare the provided APIs between MineCraft and Craftium?
> >
> > Please refer to the answer to the last weakness for a detailed comparison between Craftium and Minecraft-based alternatives. We acknowledge that the differences between Craftium and Minecraft-based alternatives were not clear in the paper. We will clarify the mentioned differences in the new version of the paper that we will upload during the next days of the rebuttal.
> >
> > [1] Baker et al. 2022. *"Video PreTraining (VPT): Learning to Act by Watching Unlabeled Online Videos"*.
> >
> > [2] Wang et al. 2023. *"Voyager: An Open-Ended Embodied Agent with Large Language Models"*.
> >
> > [3] Towers et al. 2024. *"Gymnasium: A Standard Interface for Reinforcement Learning Environments"*.

---

> ### Comment · Reviewer_TGC5 · 2024-11-22
>
> Thank you to the authors for their detailed responses.
>
> I currently have **no issue** with the contributions in terms of efficiency improvement and multi-agent support, and I believe these represent **significant and valuable contributions** to research in this field.
>
> However, some of my concerns regarding RL still remain. From what I understand, you provide only one reward function as an example in Fig. 5. Based on my interpretation, this function gives a positive reward when the agent collects a block of `tree`. Is that correct?
>
> Additionally, since Minecraft provides many command tools that allow access to game status, object counts, and block positions, I am curious: could a reward function be implemented only using these commands? If not, could you please provide more scenarios where the `Craftium` engine delivers detailed internal information about the game to compute the reward, but `Minecraft itself` does not offer commands to achieve this?
>
> By the way, when do you expect the new version of the paper to be available?
>
> If these concerns are well addressed, I would be happy to reconsider my rating and increase my confidence.
>
> Thank you in advance!

---

> > ### Author Response · Authors · 2024-11-22
> >
> > First, we sincerely appreciate your rapid response and engagement, and for recognizing the contributions of Craftium in terms of efficiency and multi-agent support. Below, we address the concerns regarding RL in Craftium.
> >
> > >  Based on my interpretation, this function gives a positive reward when the agent collects a block of tree. Is that correct?
> >
> > Yes, the reward function illustrated in Fig. 5 provides a positive reward when the agent collects a block of tree. Note that this is just one example, selected for its conciseness due to the page limitations of the paper. However many (and much more complex) reward functions can be created in Craftium. Specifically, Craftium leverages the complete [Lua programming language](https://www.lua.org/about.html) and the extensive and well-documented [Minetest API](https://api.minetest.net/), allowing to **implement practically any reward function** that RL practitioners might design.
> >
> > > [...] Minecraft provides many command tools that allow access to game status, object counts, and block positions, I am curious: could a reward function be implemented only using these commands? If not, could you please provide more scenarios where the Craftium engine delivers detailed internal information about the game to compute the reward, but Minecraft itself does not offer commands to achieve this?
> >
> > Minecraft itself does not provide any mechanism to define reward functions or integration with RL agents in general. Thus, we believe that implementing reward functions using Minecraft commands would require significant development effort and face many limitations.
> >
> > For example, commands typically require explicit invocation, which hinders their use to monitor the state of the environment and agents in real-time (i.e., in each frame or agent-environment interaction). In contrast, Craftium enables seamless and programmatic real-time access to the internal state of the engine game state via the mentioned Lua API, **a feature that Minecraft itself lacks**.
> >
> > Moreover, complex reward functions, such as those dependent on interactions between multiple agents or procedurally generated tasks, are cumbersome to implement with commands alone. Craftium leverages the Minetest API, which allows an event-driven reward computation tailored to any scenario. See, for example, the variety of [global callbacks available](https://api.minetest.net/core-namespace-reference/#global-callback-registration-functions).
> >
> > Furthermore, relying on commands to access the internal state of the engine requires additional processing overhead, such as parsing outputs or coordinating external scripts. In contrast, Craftium allows practitioners to write code (in Lua) that directly interacts with the environment's (i.e., game engine) internal state. **This includes accessing detailed information, reacting to in-game events in real-time, and defining custom behaviors or reward mechanisms efficiently and easily.**
> >
> > Consider the following example code which modifies the internal definition of spiders and spawns one in a given location.
> >
> > ```lua
> > 1 local pos = { x = 0.0, y = 4.5, z = -2.4 }
> > 2 local name = "mobs_monster:spider"
> >
> > 3 local mob_def = minetest.registered_entities[name]
> >
> > 4 mob_def.update_tag = function(self)
> > 5 end
> >
> > 6 mob_def.on_die = function(self)
> > 7   set_reward_once(1.0)
> > 8 end
> >
> > 9 minetest.add_entity(pos, name)
> > ```
> > Lines 3-8 customize the definition of a spider:
> >   + Line 3 accesses the internal definition of the spiders, i.e., the object that determines all their aspects and behavior.
> >   + Lines 4-5 replace the `update_tag` function with an empty function. Doing this prevents a text tag (infotext) from appearing displaying the monster's name in the observations of the agent every time it encounters a spider.
> >   + Lines 6-8 overwrite the `on_die` callback function of the spider, replacing it with a function that sets the reward to 1.0 for that timestep.
> > Finally, the last line spawns a spider (with the new definition) in the specified position.
> >
> > This example showcases how Craftium allows the programmatic definition of reward functions that operate in real time. The code modifies the internal definition of spiders to set the reward in real-time in response to an event (the spider dying). Reproducing this functionality using Minecraft commands would be extremely challenging or infeasible as commands cannot dynamically modify or override the behavior of entities (spiders in this case).

---

> > > ### Author Response · Authors · 2024-11-22
> > >
> > > For a much more complex example, refer to this [code](https://drive.google.com/file/d/1Xorue-ODza_mjHdzN5gsh91i_ivIleAS/view?usp=sharing) of the procedural environment generation example from Section 3.5.3 of the paper, that includes: complex automatic world generation, changing internal definitions of mobs (i.e., monsters) and items, and callbacks for real-time reward handling. For instance, real-time event-driven reward computation is showcased in lines 18-20 and 32-35. More interestingly, the code also showcases (lines 48-76) how custom 3D maps can be created using the serialized map representation described in Appendix E.3.1 of the paper.
> > >
> > > > By the way, when do you expect the new version of the paper to be available?
> > >
> > > We are currently working on the revised version of the paper, which includes multi-agent support and clarifications based on the valuable feedback we have received. We plan to upload the updated version within the next couple of days of the rebuttal period.

---

> > > ### Comment · Reviewer_TGC5 · 2024-11-24
> > >
> > > Thank you for your response. I appreciate the clarification regarding the advantage of having omniscient access to the game's internal state. This capability indeed allows for seamless computation of reward functions without requiring a separate process to call Minecraft's internal commands, which could be challenging to implement, inefficient, or even incapable of accessing certain aspects of the game's state. I am fully convinced that this framework has significant potential to advance the research and development of agents in Minecraft.
> > >
> > > However, I believe that a textual explanation alone may not be enough to convincingly demonstrate this advantage to everyone. I would highly appreciate it if the updated version of your paper includes the following:
> > >
> > > A comparison (preferably visual, such as a figure) illustrating how related works in Minecraft implement reward functions and conduct RL versus how your framework approaches these tasks.
> > > Beyond simulation speed, which you have quantitatively evaluated, could you experimentally provide evidence in other dimensions of RL (e.g., reward computation efficiency, training convergence speed, or resource consumption) to demonstrate that your framework is superior to directly using Minecraft?
> > > Additionally, I have one more question:
> > > In my humble opinion, Python is the dominant programming language in the AI community. Do you think your reliance on Lua might limit the broader adoption of your framework? If so, are there plans to mitigate this concern, such as introducing Python bindings or other integrations?
> > >
> > > Thank you for your efforts, and I look forward to your updated version!

---

> > > > ### Author Response · Authors · 2024-11-24
> > > >
> > > > First, thanks for your response and feedback. We are pleased to see that you are *"fully convinced that this framework has significant potential to advance the research and development of agents in Minecraft"*
> > > >
> > > > We have uploaded a **new version of the paper that includes multi-agent support, and further clarifications on the differences between Craftium and Minecraft-based alternatives** in the main text of the paper and in a new appendix section (Appendix G). Please refer to the general response for details, including a version of the new paper with changes in blue text.
> > > >
> > > > > [...] could you experimentally provide evidence in other dimensions of RL (e.g., reward computation efficiency, training convergence speed, or resource consumption) to demonstrate that your framework is superior to directly using Minecraft? [...]
> > > >
> > > > We would like to highlight that superior reward computation efficiency is already shown in Section 3.4 of the paper, as the evaluated environments were required to compute their reward functions. In addition to this feature and significantly improved computational efficiency compared to Minecraft-based alternatives, Craftium also supports **(1)** multi-agent environments (as outlined in the new version of the paper) and **(2)** offers near-limitless flexibility for creating environments, **two crucial features that all Minecraft-based alternatives lack**. To further clarify Craftium's advantages over MineDojo (the only Minecraft-based alternative with some environment customization capabilities), we have included a comprehensive set of examples comparing both frameworks in the newly added Appendix G.
> > > >
> > > > We hope the reviewer finds the multi-agent support, extremely improved computational efficiency, and the ability to create fully custom environments, **features that all Minecraf-based frameworks lack**, compelling evidence of Craftium's advantages over Minecraft-based alternatives.
> > > >
> > > > > Do you think your reliance on Lua might limit the broader adoption of your framework?
> > > >
> > > > We believe that the reliance on Lua does not limit Craftium's broader adoption. Lua is lightweight, widely used in game modding, and has an accessible syntax (very similar to Python's), which helps lower the barrier to adoption. This is supported by the fact that Lua is used in many extremely popular projects like Roblox, World of Warcraft, and Neovim. Additionally, users can modify existing environments through configuration options of the Python environment interface. However, this is limited compared to the Lua API, which completely eliminates the barriers between the practitioner and the underlying game engine.

---

> ### Comment · Reviewer_TGC5 · 2024-11-25
>
> I have reviewed the authors' response and the updated Appendix G.
>
> The authors highlight the following main contributions of this work:
>
> - Adding multi-intelligence support, which I find both novel and valuable.
> - Replacing the original Java-based game engine with the highly efficient Minetest, resulting in significant improvements in simulation efficiency, which I consider an important advancement.
> - Facilitating easy customization of creature attributes, behaviors, and environment generation—capabilities that are not natively available in Minecraft.
>
> Regarding the implementation of reward functions, the framework appears to be similar to [MineDojo](https://github.com/MineDojo/MineDojo/blob/main/minedojo/tasks/meta/utils/reward_fns.py), so this aspect may not constitute a major contribution.
>
> Overall, I believe this work represents a sufficient amount of workload, and adopting this framework is likely to accelerate the development of research in the relevant area.
>
> **I have accordingly upgraded my rating**.

---

### Official Review · Reviewer_CDjS · 2024-10-22

**Soundness:** 4
**Presentation:** 4
**Contribution:** 2
**Rating:** 8
**Confidence:** 4

**Summary:**

The paper introduces Craftium, a framework for creating rich 3D environments for embodied and open-ended AI research. Key contributions:
- A flexible, efficient platform built on the open-source Minetest game engine
- Easy environment creation using Lua API instead of domain-specific languages
- Significant performance improvements over Minecraft-based frameworks (+2K steps/sec)
- Integration with the Gymnasium interface for compatibility with existing RL tools
- Comprehensive examples across different use cases (classic RL, open-ended learning, continual learning)

**Strengths:**

- Built on efficient C++ codebase (Minetest) versus Java (Minecraft)
- Strong compatibility through Gymnasium interface
- Addresses real needs in embodied/open-ended AI research
- Comprehensive benchmarks against VizDoom and MineDojo
- Diverse example environments demonstrating flexibility

**Weaknesses:**

- Currently only supports single-agent scenarios
- Could use more ablation studies on design choices

The paper's claim that "Research in fields related to open-endedness... usually defaults to simplistic 2D grid environments" needs revision, as it overlooks significant recent developments in the field.

The emergence of foundation models has enabled several works to generate rich environments, such as OMNI-EPIC and EnvGen which use large language models for environment generation, but also environments such as Craftax which provides a sophisticated benchmark.

These works demonstrate that the field has already begun moving beyond simple grid-world environments. The authors should acknowledge this recent progress and better position their work within the context of these advances in environment generation for open-ended learning.

[1] OMNI-EPIC: Open-endedness via Models of human Notions of Interestingness with Environments Programmed in Code\
[2] EnvGen: Generating and Adapting Environments via LLMs for Training Embodied Agents\
[3] Craftax: A Lightning-Fast Benchmark for Open-Ended Reinforcement Learning

**Questions:**

- What are the key limitations in extending to multi-agent scenarios? Do you foresee potential challenges in extending Craftium to multi-agent scenarios, such as synchronization issues, increased computational requirements, or modifications needed to the Minetest engine.
- How does computational performance scale with environment complexity?
- What influenced the choice of Lua for environment creation?

---

> ### Author Response · Authors · 2024-11-21
>
> We thank the reviewer for the thoughtful feedback, for providing valuable references, and for the time taken to review the paper. We are pleased to announce that we have modified Craftium based on your comments to **include multi-agent support**. The following lines answer the weaknesses and questions raised by the reviewer in detail.
>
> **[Weaknesses]**
>
> > Currently only supports single-agent scenarios.
>
> Based on your feedback and similar concerns raised by other reviewers, **we have included multi-agent support in Craftium** (along with the previous single-agent capability). Multi-agent was the main line of the future work, and Craftium's code was already designed to be extended to multi-agent in the future. Now, users can select between single and multi-agent environments in Craftium. The only difference between both modes is that instead of instantiating a Minetest server and a single Minetest client (what was previously done in the single-agent version), in multi-agent, a Minetest client is run per agent. This leads to some changes in the architecture of the framework (Figure 2 of the paper), multiplying some components of the framework by the number of agents (see the new version of [Figure 2](https://drive.google.com/file/d/1Q6YICabmiXheNm0HwJh0sWcmFseIW9Va/view?usp=sharing)).
>
> To showcase this new functionality, we have included a new example environment, that consists of a one vs one combat game (screenshot available [here](https://drive.google.com/file/d/1yVrWaJu_5a4u9G9tmmKvMqIFwCsWImuT/view?usp=sharing)), where agents are rewarded (1.0) for punching other agents and penalized on damage (-0.1). In this case, we have trained the agents using self-play, a popular technique for this type of scenario (results shown [here](https://drive.google.com/file/d/1KCIOGHFhMx0EoIj5_Pk2nPoMcVlmdNAW/view?usp=sharing)) popularized by AlphaZero.  We are incorporating this new feature (and the referred figures) into a revised version of the paper that we will update during the next days of the rebuttal.
>
> With the addition of the multi-agent feature, Craftium is also the only environment creation framework that combines, fast execution speed, multi-agent, open-worlds, and procedural generation. We strongly believe that these features enhance Craftium's novelty with respect to the currently available environment frameworks.
>
> > Could use more ablation studies on design choices.
>
> Note that the paper introduces a new framework for creating environments for research in RL and autonomous agents in general and that the presented experiments are exclusively intended to showcase Craftium's capabilities. For this reason, we think that ablation studies on the employed algorithms, e.g., how different hyperparameters affect the behavior of the algorithms, don't reinforce Craftium's contributions compared to other frameworks, falling outside the scope of the paper.
>
> > [...] The authors should acknowledge this recent progress and better position their work within the context of these advances [...].
>
> Regarding the final comments, we thank the reviewer for providing new references. We will include the references in the paper and situate the work according to these.
>
> **[Questions]**
>
> > What are the key limitations in extending to multi-agent scenarios? [...]
>
> As aforementioned, we have updated Craftium to support multi-agent scenarios, and provide a new example environment, and experiments (including an updated version of Figure 2). In this process, the main challenge has been the synchronization between the environment (server) and the agents (multiple clients), which we solved using already existing functionalities of Minetest and POSIX.
>
> > How does computational performance scale with environment complexity?
>
> Note that the metrics shown in Figure 7 aggregate results from Craftium environments of different complexity. The results show low variance in the steps per second metric (note the small black line above Craftium's bar in the plot). Therefore, we can conclude that environment complexity does not heavily affect performance.
>
> > What influenced the choice of Lua for environment creation?
>
> The usage of Lua for creating environments in Craftium is completely influenced by Minetest, as it is the language Minetest employs to extend and modify the behavior of the game engine. Furthermore, Lua has many additional benefits: being very similar to well-known and simple programming languages like Python, very fast compared to other interpreted programming languages, and having extensive support and community.

---

> ### Comment · Reviewer_CDjS · 2024-11-23
>
> Thank you for your comprehensive response and the significant improvements made to Craftium.
>
> Research in reinforcement learning and open-endedness suffers from too simple environments. While pure GPU-accelerated environments can achieve higher throughput, Craftium stands out as the fastest Minecraft-like plaftform currently available for research. This is a significant achievement given the inherent complexity of voxel-based worlds and the rich interaction possibilities they offer. In the end, I think that Craftium is a valuable contribution to the community.
>
> Thank you for addressing all my concerns. I look forward to reading the revised manuscript.

---

> > ### Author Response · Authors · 2024-11-24
> >
> > First, we sincerely appreciate the reviewer's kind words. We are glad to hear that the raised concerns have been successfully addressed. We have uploaded a revised version of the paper, which includes the reviewer's feedback and the added multi-agent support. Please refer to the general response for details, including a version of the new paper with changes in blue text. We believe these changes further strengthen the paper's contributions, and we hope the reviewer finds them satisfactory.
> >
> > Again, thanks for your response and the kind words.

---

### Official Review · Reviewer_mCbn · 2024-11-04

**Soundness:** 3
**Presentation:** 2
**Contribution:** 2
**Rating:** 3
**Confidence:** 4

**Summary:**

The paper presents Craftium, a framework designed to provide computationally efficient and customizable 3D environments for open-ended and embodied AI research, using the Minetest engine. Craftium aims to address limitations in existing environments, which are either computationally demanding or lack flexibility. The authors highlight its compatibility with the Gymnasium API, ease of use in procedural generation, and increased performance over other platforms like Minecraft.

**Strengths:**

- Craftium provides a complex 3D environment similar to Minecraft but offers faster performance, achieving over 2,000 steps per second compared to Minecraft-based frameworks.
- It allows for efficient procedural generation and supports a variety of reinforcement learning (RL) tasks.

**Weaknesses:**

- The paper lacks a detailed comparison with other widely used simulators in robotics, such as IsaacGym, PyBullet, or MuJoCo, which also support 3D environments and open-world scenarios.
- The experimental comparisons are weak, and details on baseline and alternative environment performance are limited. The paper doesn’t clearly outline the real-world application of its contributions or benchmarks for similar environment capabilities in existing platforms.
- Overall, the novelty of the framework is minimal, as the setup resembles that of existing environments, with modifications focused on efficiency rather than new functionality.
- In the procedural environment generation section, experiments demonstrated the generation of environments. However, all the generated environments can be represented using ASCII, meaning they are 2D environments. What is shown is a 3D representation of a 2D environment.
- In table 1, it is difficult to evaluate exactly if a framework is active or not after some time. There is no guarantee that this framework will still be active after a few months. Hence, I think that it is not a comparable characteristic.
- Missing related works that show the possibility of creating 3D environments in an open-ended way [1, 2, 3].

[1] Lee, J., Hwangbo, J., Wellhausen, L., Koltun, V., & Hutter, M. (2020). Learning quadrupedal locomotion over challenging terrain. Science robotics, 5(47), eabc5986.
[2] Faldor, M., Zhang, J., Cully, A., & Clune, J. (2024). OMNI-EPIC: Open-endedness via Models of human Notions of Interestingness with Environments Programmed in Code. arXiv preprint arXiv:2405.15568.
[3] Earle, S., Kokkinos, F., Nie, Y., Togelius, J., & Raileanu, R. (2024, May). Dreamcraft: Text-guided generation of functional 3D environments in Minecraft. In Proceedings of the 19th International Conference on the Foundations of Digital Games (pp. 1-15).

**Questions:**

- How does Craftium's procedural generation approach compare in diversity and scalability to existing frameworks in continuous and open-ended RL settings?
- Why is “spiders attack” a challenging task? It seems to have denser reward signals than “chop tree”.
- How do training of the tasks compare with that of existing simulators (e.g., MineDojo)? What is the performance against real time?
- In Figure 11, why are there 2 maximum red lines for the LLava-Agent? Why is it in % of steps, and not absolute number of steps?
- In Figure 12, why did the agent not manage to get any reward in the first 2 generated environments? They look like the simplest environments. In any of the generated environments, did the agent successfully complete the task of reaching the diamond?

---

> ### Author Response · Authors · 2024-11-21
>
> First, we thank the reviewer for the detailed comments and thorough feedback which helped improve the paper. In the following lines, we address the weaknesses and questions raised by the reviewer.
>
> **[Weaknesses]**
>
> > The paper lacks a detailed comparison with other widely used simulators in robotics, such as IsaacGym, PyBullet, or MuJoCo [...]
>
> We would like the reviewer to note the substantial difference between the focus of the simulators that the reviewer refers to and the target research area that Craftium contributes to. IsaacGym, PyBullet, and MuJoCo are simulators mainly used for research in robotics, focusing on low-level control tasks. Thus, these simulators are specialized in precisely modeling real-world physics dynamics. In contrast, Craftium, like MineDojo, MiniHack, and VizDoom, contributes to other areas of RL and autonomous agent research, targeting high-level cognitive challenges such as planning [1], exploration [2], long-term reasoning [3]. Having such different research targets and goals makes Craftium and the other listed environments focus on very different features when compared to physics simulators like IsaacGym, PyBullet, and MuJoCo: large open worlds, procedural generation, and environmental richness (e.g., diverse biomes, weather systems, animals, and other entities). Therefore, Craftium should be compared to other frameworks with a similar purpose, such as MineDojo and VizDoom, as discussed in the paper. Within this context, Craftium represents significant novelty (ability to create completely custom environments, flexibility, extremely fast execution speed, multi-agent support, etc.), as further elaborated in the answer to the next weakness.
>
> Finally, the reviewer states that IsaacGym, PyBullet, and MuJoCo *"support 3D environments and open-world scenarios"*. Although we completely agree on the ability to create 3D environments, to the best of our knowledge, these frameworks do not currently support open-world scenarios, although we are open to discussion. Note that we understand the definition of an open world as described [here](https://en.wikipedia.org/wiki/Open_world).
>
> > [...] the novelty of the framework is minimal, as the setup resembles that of existing environments [...]
>
> We want to emphasize that Craftium is not just a set of predefined environments but a framework for **creating completely new and custom environments**. While Section 3.5 includes example environments, they are only provided to showcase Craftium's capabilities and serve as a starting point for users to create their environments.
> When compared to other works that share the same objectives and target similar research (MineDojo, MineRL, MiniHack, NLE, VizDoom, etc.), Craftium offers significant innovations:
> - Craftium is the only framework with the richness of Minecraft-like environments and exceptional computational efficiency. Efficiency should not be trivialized, as it is often a critical factor for enabling running experiments for most research labs. Low computational cost also translates into democratizing access to technology and research.
> - Craftium is extremely versatile and easy to use compared to alternatives. The ability to use an extremely flexible (and thoughtfully documented) Lua API to design rich 3D environments is a novel feature considering other environment creation frameworks (see lines 71–82 and 472–490 in the paper).
> - With the **recent addition of multi-agent support** (details in the general answer above), to the best of our knowledge, Craftium is the only 3D environment creation framework that combines open-worlds, procedural generation, and the richness of Minecraft with multi-agent capabilities.
>
> > [...] the generated environments can be represented using ASCII, meaning they are 2D environments.
>
> Note that the fact that these environments can be represented in ASCII does not imply that the generated environments are 2D. Although ASCII is used to serialize the environments, the ASCII format (as explained in Appendix E.3.1) defines the environment's map in vertical layers, resulting in a 3D map.
>
> > [...] it is difficult to evaluate exactly if a framework is active or not after some time [...] I think that it is not a comparable characteristic.
>
> We agree that determining whether a framework is active or not can sometimes be unclear. However, we believe this is a critical factor, as inactive frameworks often present significant limitations for integration into new research projects. To reduce ambiguity, we explicitly define *inactive* in the paper (see line 435) as frameworks without updates for over a year. Additionally, some frameworks are officially declared unmaintained by their authors, further supporting this classification. Finally, our future research makes extensive use of Craftium, as such, maintaining Craftium in the long term is crucial for us.

---

> > ### Author Response · Authors · 2024-11-21
> >
> > > [...] Missing related works [...]
> >
> > We truly appreciate the provided references and will include them in the revised version of the paper, which we will upload in the rebuttal period.
> >
> > **[Questions]**
> >
> > > How does Craftium's procedural generation approach compare in diversity and scalability to existing frameworks in continuous and open-ended RL settings?
> >
> > When the reviewer refers to *"continuous open-ended RL settings"*, we want to clarify that we understand the mentioned setting as open-ended unsupervised environment design (UED) using continuous control scenarios, similar to the [POET paper by Wang et al. 2019](https://arxiv.org/abs/1901.01753). In this case, note that Craftium is not an approach for UED, but it is a framework for generating environments. In this sense, Craftium is very valuable for UED research, as it opens the door to many new directions in which modify and generate environments in an open-ended way compared to scenarios based on simulators like MuJoCo: different [biomes](https://wiki.minetest.net/Biomes), [animals](https://content.luanti.org/packages/TenPlus1/mobs_animal/), [monsters](https://content.luanti.org/packages/TenPlus1/mobs_monster/), weather, day/night cycles, a wide range of [items](https://wiki.minetest.net/Games/Minetest_Game/Items) or complex interactions with the environment (building, destructing, gathering resources, etc.).
> >
> > > Why is “spiders attack” a challenging task? It seems to have denser reward signals than “chop tree”.
> >
> > We refer to spiders attacks as more challenging as having multiple enemies attacking simultaneously from various directions (in the case of the spiders) makes the environment significantly more challenging to solve (solving would correspond to defeating all spiders in the last round) compared to chopping trees. As can be seen in the values episodic returns from Figure 8, the agent can chop +6 (+10 sometimes) trees, while the low episodic return value in the spiders attack environment corresponds to only defeating a single spider half of the times (see lines 950-956 in Appendix E.1).
> >
> > > How do training of the tasks compare with that of existing simulators (e.g., MineDojo)? What is the performance against real time?
> >
> > The training of the tasks is very similar in both frameworks, as for training agents, the user handles the environment via the Gymnasium API, in the case of Craftium, and via Gym (the deprecated library that continues its development in Gymnasium) in the case of MineDojo. However, as demonstrated in Section 3.4, Craftium environments are significantly more efficient than MineDojo's, resulting in much faster training times, reducing the time between development and testing, and increasing the productivity of the researcher.
> >
> > Regarding the second part of the question, considering 60FPS (or interactions per second) as real-time (60FPS is the most common frame rate), Craftium environments can be run at thousands of FPS, greatly reducing the time and financial cost of the experiments.
> >
> > > In Figure 11, why are there 2 maximum red lines for the LLava-Agent? Why is it in % of steps, and not absolute number of steps?
> >
> > The dashed line corresponds to the average (across different runs) maximum episodic return in the current timestep. The solid line refers to the best episodic return value obtained across all runs. The X-axis is represented as the percentage of the time of the agent in the environment. This is a consequence of the fact that the timesteps of the PPO+LSTM agent do not correspond to the interactions of the LLaVa-Agent with the environment.
> >
> > > In Figure 12, why did the agent not manage to get any reward in the first 2 generated environments? [...] did the agent successfully complete the task of reaching the diamond?
> >
> > We believe that the reason behind the low reward in the first two tasks corresponds to the fact that the agent has not only to reach the diamond but has to collect it. Collecting it is more challenging as it requires the agent to center the camera view on the diamond and take the action to collect it. Further in the task sequence, the agent successfully collects the diamond sometimes (e.g., in task 3).
> >
> > [1] Hafner et al., 2023. *"Mastering Diverse Domains through World Models"*.
> >
> > [2] Ermolov et al., 2021. *"First Return, Then Explore"*.
> >
> > [3] Wang et al., 2023. *"Voyager: An Open-Ended Embodied Agent with Large Language Models"*.

---

> > > ### Comment · Reviewer_mCbn · 2024-11-23
> > >
> > > I appreciate the authors' clarifications, and the additional features of multi-agent support presented. While I acknowledge the potential of Craftium, I find that the experiments (or newly included figures) do not provide sufficiently convincing evidence to support the central claim that Craftium can generate rich and diverse environments.
> > >
> > > > IsaacGym, PyBullet, and MuJoCo are simulators mainly used for research in robotics, focusing on low-level control tasks. Thus, these simulators are specialized in precisely modeling real-world physics dynamics. In contrast, Craftium, like MineDojo, MiniHack, and VizDoom, contributes to other areas of RL and autonomous agent research, targeting high-level cognitive challenges such as planning [1], exploration [2], long-term reasoning [3].
> > >
> > > Thank you for clarifying the distinction between Craftium and traditional robotic simulators such as IsaacGym, PyBullet, and MuJoCo. I understand the emphasis on high-level cognitive challenges that Craftium aims to address. However, given this focus, I believe it would be important to include comparisons with other embodied simulators that are often utilized for these purposes, such as Habitat, AI2-THOR, and ProcTHOR.
> > >
> > > Additionally, I would like to point out that it is not entirely accurate to suggest that IsaacGym, PyBullet, or MuJoCo cannot address cognitive challenges. For example, OMNI-EPIC has demonstrated the feasibility of using a simplified robotic model to create diverse environments within PyBullet to tackle such tasks. While I recognize the difference in contributions of Craftium, I believe the writing could be improved to better clarify these distinctions.
> > >
> > > > Craftium is extremely versatile and easy to use compared to alternatives. The ability to use an extremely flexible (and thoughtfully documented) Lua API to design rich 3D environments is a novel feature considering other environment creation frameworks (see lines 71–82 and 472–490 in the paper).
> > >
> > > I appreciate the emphasis on the versatility and ease of use of Craftium, as well as the flexibility of its Lua API for designing rich 3D environments. However, I remain unconvinced that the paper provides sufficient evidence to support this claim. As mentioned in my earlier comment, the environments shown in Figure 12 appear to be primarily 3D representations of 2D environments. To adequately demonstrate this point, it would be necessary to showcase the generation (or procedural generation) of genuinely rich and complex 3D environments using Craftium.
> > >
> > > > We agree that determining whether a framework is active or not can sometimes be unclear. However, we believe this is a critical factor, as inactive frameworks often present significant limitations for integration into new research projects. To reduce ambiguity, we explicitly define inactive in the paper (see line 435) as frameworks without updates for over a year.
> > >
> > > I agree that having active repositories and up-to-date code is highly beneficial. However, I remain unconvinced that this should be considered a scientific measure in a paper, as the activity status of a framework is a dynamic characteristic that can change over time. While it is useful to note, it may not serve as a reliable criterion for evaluation.
> > >
> > > >  In this sense, Craftium is very valuable for UED research, as it opens the door to many new directions in which modify and generate environments in an open-ended way compared to scenarios based on simulators like MuJoCo: different biomes, animals, monsters, weather, day/night cycles, a wide range of items or complex interactions with the environment (building, destructing, gathering resources, etc.).
> > >
> > > I agree that a computationally efficient simulator would be highly valuable for UED research, particularly one that supports diverse features such as different biomes, animals, weather, and complex interactions with the environment. However, based on the experiments presented, I remain unconvinced that the environments generated are sufficiently diverse or computationally efficient. As mentioned earlier, demonstrating the ability to easily generate rich and complex 3D environments with Craftium would strengthen this claim.
> > >
> > > > We believe that the reason behind the low reward in the first two tasks corresponds to the fact that the agent has not only to reach the diamond but has to collect it. Collecting it is more challenging as it requires the agent to center the camera view on the diamond and take the action to collect it. Further in the task sequence, the agent successfully collects the diamond sometimes (e.g., in task 3).
> > >
> > > If the agent could not collect the diamond at an easier level, how could it collect the diamond at a more difficult level?
> > >
> > > > added support for multi-agent scenarios
> > >
> > > This is a great feature! However, since there are no extensive experiments or revisions to the manuscript yet, I am unable to provide any feedback on it at this time.

---

> > > > ### Author Response · Authors · 2024-11-24
> > > >
> > > > We thank the reviewer for the feedback, references, and engagement. First, note that we have updated the paper with a new version including your feedback, new references, and multi-agent support. Please refer to the general response above for details. In the following lines we summarize how the raised concerns have been addressed.
> > > >
> > > > > I find that the experiments (or newly included figures) do not provide sufficiently convincing evidence to support the central claim that Craftium can generate rich and diverse environments.
> > > >
> > > > Note that Craftium builds on top of the Minetest game engine and that it exposes Minetest's complete [Lua API](https://api.minetest.net/) to the practitioner to develop environments in Craftium. Therefore, Craftium's abilities to create diverse and rich environments are those of Minetest. Examples of such richness and diversity are the environments (games) that have been created in Minetest. Please refer to [VoxeLibre](https://content.luanti.org/packages/Wuzzy/mineclone2/), [Mineclonia](https://content.luanti.org/packages/ryvnf/mineclonia/) for specific examples. Note that these are all developed using **exactly** the same Lua API that is used in Craftium to develop environments. Moreover, these examples are integrated seamlessly in Craftium, as is the case of the open-world environment from Section 3.5.3 of the new version of the paper. In this section, the [VoxeLibre](https://content.luanti.org/packages/Wuzzy/mineclone2/) game is used as the basis for implementing the open-world environment. Additionally, Figure 10 shows examples of the richness and diversity of the generated environment. We believe that the latter constitutes a strong evidence of the possibility of generating rich and diverse environments in Craftium.
> > > >
> > > > > I believe it would be important to include comparisons with other embodied simulators that are often utilized for these purposes, such as Habitat, AI2-THOR, and ProcTHOR. [...] For example, OMNI-EPIC [...] I believe the writing could be improved to better clarify these distinctions.
> > > >
> > > > Thanks for the feedback and new references, we have updated the paper accordingly. The new introduction section incorporates Habitat 3.0 and OMNI-EPIC references, and Section 4 and Table 1 explicitly compare Craftium with Habitat 3.0 and AI2-THOR. Regarding ProcTHOR, to the best of our knowledge, it is based on AI2-THOR and only enhances its procedural generation capabilities, thus, we have only included AI2-THOR in Section 4 and considered ProcTHOR as part of its ecosystem, as they have very similar characteristics. Please correct us if this is not the case.
> > > >
> > > > > To adequately demonstrate this point, it would be necessary to showcase the generation (or procedural generation) of genuinely rich and complex 3D environments using Craftium.
> > > >
> > > > Note that Section 3.5.4 is **just one example** of the environments that can be procedurally generated in Craftium. Please refer to Section 3.5.3 of the new version of the paper which also showcases **exactly this feature**. This section employs [VoxeLibre](https://content.luanti.org/packages/Wuzzy/mineclone2/) as the basis for creating a **procedurally generated** open-world environment of great richness and diversity. Please refer to Figure 10 of the paper, as it demonstrates the procedural, rich, and complex scenarios that Craftium allows.
> > > >
> > > > > I agree that having active repositories and up-to-date code is highly beneficial. However, I remain unconvinced that this should be considered a scientific measure in a paper, as the activity status of a framework is a dynamic characteristic that can change over time. While it is useful to note, it may not serve as a reliable criterion for evaluation
> > > >
> > > > The new version of the paper replaces the project status column with support for multi-agent scenarios and open worlds. Please refer to the new version of the paper for details.
> > > >
> > > > > If the agent could not collect the diamond at an easier level, how could it collect the diamond at a more difficult level?
> > > >
> > > > We believe that the agent collects the diamond in the latter tasks because it has learned to defend itself from monsters, as defending from monsters (using the sword) is the same action as collecting the diamond.

---

> > > > > ### Author Response · Authors · 2024-11-24
> > > > >
> > > > > > This is a great feature! However, since there are no extensive experiments or revisions to the manuscript yet, I am unable to provide any feedback on it at this time.
> > > > >
> > > > > We appreciate the reviewer's comment on the multi-agent support as a valuable feature. We have updated the paper to include multi-agent support, along with the reviewer’s feedback, mainly: we have incorporated more related works in the intro, add a new section (Section 3.5.2), updated figures, new figures with experimental results, modifications to Section 4 and Table 1 to include the mentioned simulators, and new appendix sections (Appendix D, F.2, and G).  Please refer to the general response for details, including a version of the new paper with changes in blue text. We believe these additions will strengthen the paper and encourage the reviewer to reconsider the score.

---

> > > > > > ### Comment · Reviewer_mCbn · 2024-11-25
> > > > > >
> > > > > > Thank you for the additional clarifications. I have updated my score accordingly.

---

### Official Review · Reviewer_Bq4e · 2024-11-04

**Soundness:** 4
**Presentation:** 3
**Contribution:** 3
**Rating:** 8
**Confidence:** 4

**Summary:**

The paper introduces Craftium, a new platform leveraging the open-source Minetest game engine to enable the creation of customizable, efficient, and rich 3D environments for research in RL and open-ended agent scenarios. Craftium is designed to bridge the gap between more simple 2D environments and computationally demanding, closed-source 3D platforms like Minecraft. Through various benchmark tests, the authors demonstrate that Craftium achieves significant performance gains, being notable faster than the Minecraft-based frameworks, and present use cases illustrating its flexibility for diverse RL tasks, procedural environment generation, and complex open-world simulations.

Minor comments:
- There is a typo on line 1027 “Then, the agent that the agent, …”
- This paper seems like relevant work: Grbic, Djordje, et al. "Evocraft: A new challenge for open-endedness." Applications of Evolutionary Computation: 24th International Conference, EvoApplications 2021.

**Strengths:**

- Craftium addresses a critical limitation in AI research, especially in RL and embodied AI, by enabling complex and computationally feasible 3D environments that are both customizable and efficient.

- The paper presents comprehensive benchmark comparisons, e.g. MineDojo and VizDoom, demonstrating its advantage over current platforms in terms of computational performance and versatility.

- The authors provide clear explanations of Craftium’s architecture, including how Minetest’s open-source flexibility allows for modifications crucial for RL, such as reward functions and synchronous client-server interactions.

**Weaknesses:**

The framework currently supports single-agent settings, with multi-agent scenarios identified as future work. Given the growing interest in multi-agent RL, this limitation restricts Craftium’s immediate applicability for research in cooperative and competitive agent settings.

While the paper makes a valuable contribution to the field of environment design for embodied and open-ended AI research, the framework is basically "just" a wrapper around Minetest, thus limiting its novelty.

**Questions:**

Does the current framework support Minecraft Redstone components? These would be useful to allow agents to build more complex structures.

---

> ### Author Response · Authors · 2024-11-21
>
> First, we thank the reviewer for the feedback and comments that have improved the paper. Regarding the comments, we have modified the paper to fix the typo and include the provided reference. In the following lines, we answer the raised weaknesses and questions.
>
> **[Weaknesses]**
>
> > The framework currently supports single-agent settings [...] this limitation restricts Craftium’s immediate applicability for research in cooperative and competitive agent settings.
>
> We completely agree. In response to what seems to be the major weakness of our work, we are pleased to announce that **we have updated Craftium to also support multi-agent environments**. Multi-agent was the main line of the future work, and Craftium's code was already designed to be extended to multi-agent in the future. Now, users can select between single and multi-agent environments in Craftium. The only difference between both modes is that instead of instantiating a Minetest server and a single Minetest client (what was previously done in the single-agent version), in multi-agent, a Minetest client is run per agent. This leads to some changes in the architecture of the framework (Figure 2 of the paper), multiplying some components of the framework by the number of agents (see the new version of [Figure 2](https://drive.google.com/file/d/1Q6YICabmiXheNm0HwJh0sWcmFseIW9Va/view?usp=sharing)).
>
> To showcase this new functionality, we have included a new example environment, that consists of a one vs one combat game (screenshot available [here](https://drive.google.com/file/d/1yVrWaJu_5a4u9G9tmmKvMqIFwCsWImuT/view?usp=sharing)), where agents are rewarded (1.0) for punching other agents and penalized on damage (-0.1). In this case, we have trained the agents using self-play, a popular technique for this type of scenario (results shown [here](https://drive.google.com/file/d/1KCIOGHFhMx0EoIj5_Pk2nPoMcVlmdNAW/view?usp=sharing)) popularized by AlphaZero.  We are incorporating this new feature (and the referred figures) into a revised version of the paper that we will update during the next days of the rebuttal.
>
> With the addition of the multi-agent feature, Craftium is also the only environment creation framework that combines, fast execution speed, multi-agent, open-worlds, and procedural generation. We strongly believe that these features make Craftium a substantially novel framework with respect to the currently available environment frameworks and that it is a significant contribution to the field of RL and autonomous agents in general.
>
> > While the paper makes a valuable contribution to the field of environment design for embodied and open-ended AI research, the framework is basically "just" a wrapper around Minetest, thus limiting its novelty.
>
> We are very pleased to see that the reviewer considers Craftium *"a valuable contribution to the field"*. However, we believe that Craftium goes beyond being a wrapper around Minetest. First, as the reviewer notes we have included *"modifications crucial for RL"* in Minetest. Note that these modifications (sending and receiving data to/from Python processes extremely fast, client/server synchronization, extensions to the Lua API, complex inventory interactions, etc.) are non-trivial, as Minetest is a complex game engine. Furthermore, besides the library, Craftium is comprised of ready-to-use (currently eight) environments that cover a wide range of use cases, scripts for training RL agents, integration with LMM agents, extensive documentation (with examples, API documentation, tutorials, etc.), and more.
>
> **[Question]**
>
> > Does the current framework support Minecraft Redstone components? These would be useful to allow agents to build more complex structures.
>
> Although Minecraft's Redstone is not directly implemented, Minetest (and consequently Craftium) counts with the [Mesecons](https://mesecons.net/) mod, which includes features very similar to Redstone. From the Mesecons webpage:
>
> > *"If you already know about Minecraft's redstone, you will also be familiar with Mesecons. Mesecons are not just redstone though - they share some similarities, but are also way more powerful than redstone thanks to programmable blocks such as the Luacontroller, logic gates and different kinds of sensors."*

---

> > ### Author Response · Authors · 2024-11-24
> >
> > We are pleased to announce that we have updated the paper to include multi-agent support and incorporate your feedback, mainly: a new section (Section 3.5.2), updated figures, new figures with experimental results, new appendix sections (Appendix D and F.2), and new references on the topic. Please refer to the general response for details, including a version of the new paper with changes in blue text. We strongly believe that these changes have significantly improved our work and greatly enhanced its contributions and novelty. We hope the reviewer finds the new version paper satisfactory.

---

> > ### Comment · Reviewer_Bq4e · 2024-11-26
> >
> > Thank you for the clarifications. I agree, the multi-agent support is great and definitely increases the usefulness of the system.
> >
> > In that context, I was wondering if the current version also supports training through an evolutionary algorithm? You mention a few times that the modification are crucial for RL but this environment could also be very useful for experiments in neuroevolution.

---

> > > ### Author Response · Authors · 2024-11-26
> > >
> > > We are sincerely grateful to the reviewer for the time dedicated to reviewing our paper and engaging in the rebuttal.
> > >
> > > > I was wondering if the current version also supports training through an evolutionary algorithm?
> > >
> > > Yes, Craftium's interface is designed to be general and supports training through evolutionary algorithms and practically any other learning method. For example, Section 3.5.3 describes the use of a large multi-model model agent (LLaVA-Agent) that does not rely on RL, illustrating the framework's versatility.
> > >
> > > For clarity, we have included additional clarifications in a new version of the paper. Specifically, changes include a footnote in the introduction section (footnote 2) and lines 424-425 in Section 3.5.3 to explicitly mention that Craftium can be employed beyond RL to other learning paradigms like evolutionary algorithms.
> > >
> > > We are pleased that you find the newly added multi-agent support a valuable feature that *"definitely increases the usefulness of the system"*. We hope that the new version of the paper including multi-agent support, along with Craftium's flexibility to integrate different learning paradigms and the corresponding clarifications in the paper are enough contributions to encourage the reviewer to reconsider the provided score.
> > >
> > > Again, thank you for the dedicated time.

---

> > > > ### Comment · Reviewer_Bq4e · 2024-11-29
> > > >
> > > > Thank you! Given those additions, I'm happy to raise my score.

---

### Author Response · Authors · 2024-11-21

First, we thank the reviewers for dedicating their time and effort to read our paper and provide thoughtful feedback.

**Summary:** We are pleased to see that reviewers stated that *"Craftium addresses a critical limitation in AI research"* (**R-Bq4e**), *"Addresses real needs in embodied/open-ended AI research"* (**R-CDjS**), and features *"diverse examples demonstrating its flexibility"* (**R-CDjS**). Reviewers also highlight that the paper is well-written (**R-TGC5**).

At the same time, two main concerns were raised regarding the **(1)** lack of multi-agent support, **R-Bq4e** and **R-CDjS**, and **(2)** the novelty of our contributions, **R-mCbn** and **R-TGC5**. We are happy to announce that **we have added multi-agent support to Craftium**. This new feature directly solves the first concern and significantly enhances the novelty of Craftium, **addressing both issues**. Below we summarize how these concerns were addressed.

**(1) Multi-agent support.**

We have incorporated multi-agent support into Craftium. This update was facilitated by Minetest's existing multiplayer capabilities and Craftium's initial design, built with a future multi-agent extension in mind.

Previously Craftium executed its environments by running Minetest in single-client mode, where the server and the client were handled by the same process. Now, Craftium can run Minetest in dedicated server mode, allowing multiple clients to connect (see the updated [Figure 2](https://drive.google.com/file/d/1Q6YICabmiXheNm0HwJh0sWcmFseIW9Va/view?usp=sharing)). This feature enables instantiating a Minetest client per agent, that connects to the same server, effectively sharing the same environment. Importantly, this new feature is an addition, that preserves all previous features including single-agent mode, ease of use, procedural generation, versatility, open-world capabilities, computational efficiency, etc.

To illustrate the new capability, we have created an example multi-agent environment (see [screenshot](https://drive.google.com/file/d/1yVrWaJu_5a4u9G9tmmKvMqIFwCsWImuT/view?usp=sharing)). The environment is a one vs one multi-agent combat game, where agents are rewarded (+1) for punching other agents and penalized (-0.1) on damage. We have trained the agents using self-play, a popular approach for this type of scenario [1]. As can be seen in this [plot](https://drive.google.com/file/d/1KCIOGHFhMx0EoIj5_Pk2nPoMcVlmdNAW/view?usp=sharing), the agents can successfully learn the proposed task, showcasing the multi-agent research possibilities of the framework. In the following days, we will upload a revised version of the paper version including the multi-agent feature as well as the new environment and experiments.

---

> ### Author Response · Authors · 2024-11-21
>
> **(2) Novelty.**
>
> Regarding the second concern, multi-agent support significantly enhances Craftium's novelty, as it is the **only** 3D environment creation framework that combines the richness of Minecraft environments (many [biomes](https://wiki.minetest.net/Biomes), [animals](https://content.luanti.org/packages/TenPlus1/mobs_animal/), [monsters](https://content.luanti.org/packages/TenPlus1/mobs_monster/), weather, day/night cycles, a wide range of [items](https://wiki.minetest.net/Games/Minetest_Game/Items), etc.) with multi-agent support, opening the door to many new possibilities in the RL, open-ended, and embodied AI research fields.
>
> Beyond multi-agent support, **R-mCbn** and **R-TGC5** expressed concerns about its contribution relative to existing environments.
>
> **R-mCbn** raised concerns about the novelty of Craftium regarding physics simulators like IsaacGym, PyBullet, and MuJoCo, which **follow very different objectives** compared to Craftium. These simulators are specifically designed for RL research in **robotics**, focusing on low-level control tasks that require precise physical modeling of real-world dynamics. From the MuJoCo's webpage:
>
> > *"MuJoCo is a free and open source physics engine that aims to facilitate research and development in robotics, biomechanics, graphics and animation, and other areas where fast and accurate simulation is needed."*
>
> In contrast, Craftium and other popular works such as MineRL, MineDojo, MiniHack, Atari, NLE, VizDoom, and more, **target a different domain**: high-level cognitive challenges such as planning [2], exploration [3], and long-term reasoning [4]. We believe that comparing frameworks like Craftium or MineDojo to physics simulators is outside the scope of the paper, as they address very different research objectives and require other features. For example, Craftium and MiniHack feature procedurally generated open worlds, which is infeasible in the mentioned simulators prioritizing accurate physical modeling.
>
> **R-mCbn** raised some questions about Craftium's novelty compared to Minecraft. Craftium offers significant improvements to Minecraft-based environments, such as the ability to create completely custom environments, greatly improved computational efficiency, multi-agent support, and being completely open source.
>
> We will include the mentioned clarifications on Craftium's novelty in the new version of the paper that we will upload during the next days of the rebuttal.
>
> Below, we answer the weaknesses and questions of the individual reviewers.
>
> [1] Silver et al., 2017. *"Mastering Chess and Shogi by Self-Play with a General Reinforcement Learning Algorithm"*
>
> [2] Hafner et al., 2023. *"Mastering Diverse Domains through World Models"*.
>
> [3] Ermolov et al., 2021. *"First Return, Then Explore"*.
>
> [4] Wang et al., 2023. *"Voyager: An Open-Ended Embodied Agent with Large Language Models"*.

---

> > ### Author Response · Authors · 2024-11-24
> > **Paper update**
> >
> > We are pleased to announce that we have updated the paper to address the reviewers' feedback. Key improvements include the **addition of the multi-agent feature, enhanced related work and references, and numerous examples demonstrating Craftium's versatility in creating new environments compared to Minecraft-based alternatives**. For clarity, a version of the paper highlighting changes in blue text is available **[here](https://drive.google.com/file/d/1B4R49kVSelE73ONtv0i8-3vheyT7ySbr/view?usp=sharing)**. Below is a summary of the modifications.
> >
> > - The title is now: *"Craftium: A Framework for Creating Single and Multi-Agent Environments for Open-Ended and Embodied AI"*
> > - Changes to the text of the whole paper to incorporate the multi-agent feature, mainly: a new section (Section 3.5.2), updated figures, new figures with experimental results, new appendix sections (Appendix D and F.2), modifications to Section 4 and Table 1, and new references on the topic.
> > - New references include: EvoGen, Dreamcraft, OMNI-EPIC; and new environment frameworks: Craftax, AI2-THOR, and Habitat 3.0.
> > - New version of Figure 2 to incorporate the multi-agent setting, with the corresponding text in Section 3 (lines 147-156).
> > - Incorporate PettingZoo support (multi-agent interface similar to Gymnasium) in the paper, mostly in Section 3.3.
> > - New section to hold the new multi-agent environment example and corresponding experiments: "Example 2: Multi-Agent Reinforcement Learning" and Figure 9 showing the learning curves.
> > - Introduce new frameworks in Table 1: AI2-THOR, Habitat 3.0, and Craftax.
> > - Modify Section 4 to include the newly incorporated frameworks (lines 496-508).
> > - New Appendix F.2 to describe the multi-agent environment and experiment in detail.
> > - New Appendix D to show the usage of PettingZoo with Craftium.
> > - New Appendix G to showcase the environment creation capabilities of Craftium compared to Minecraft-based frameworks.
> > - Minor fixes and typo corrections.
> >
> > These modifications address the multi-agent support concerns raised by **R-Bq4e** and **R-CDjS**, as well as incorporate feedback on frameworks for robotic embodied AI from **R-mCbn** in the revised introduction, Table 1, and Section 4. Additionally, we have clarified the differences between Minecraft-based environments and Craftium and included a new appendix (Appendix G) with detailed examples on this topic, addressing the concern of **R-TGC5**. Overall, we believe these changes significantly enhance Craftium's contribution to the field and hope the reviewers will consider revising their scores accordingly. Individual responses to the reviewers are provided below.

---

### Meta-Review · Area_Chair_tUk5 · 2024-12-19

**Metareview:**

In its current state, Craftium is an impressive engineering contribution, but the paper, in its current state, does not provide strong evidence supporting its motivation to provide a rich, open-ended environment simulator that provides benefits over preexisting environments. Undoubtedly, Minecraft is a powerful system with the potential to provide this, but the current experiments focus almost entirely on environments whose dynamics are approximately the same as offered in commonly used environments such as NLE/MiniHack, Crafter/Craftax, MuJoCo, and other versions of Minecraft for RL (as pointed out by reviewer mCbn and TGC5). Therefore, the primary contribution demonstrated in this paper is a more efficient wrapper of Minecraft. I encourage the authors to refocus their experiment design to truly demonstrate the claimed benefits of their implementation of Minecraft.

**Additional Comments On Reviewer Discussion:**

Many reviewers suggested incorporating the multi-agent setting, however, it is not clear what benefits and challenges this toy multi-agent (1v1) environment provides over other multi-agent environments that already exist.

---

### Decision · Program_Chairs · 2025-01-22

Reject